# Physiological and Molecular Analysis Revealed the Role of Silicon in Modulating Salinity Stress in Mung Bean

**Musa Al Murad** [1,2] **and Sowbiya Muneer** [1,*]

1 Horticulture and Molecular Physiology Lab, School of Agricultural Innovations and Advanced Learning, Vellore Institute of Technology, Vellore 632014, Tamil Nadu, India; almuradmusa@gmail.com
2 School of Biosciences and Technology, Vellore Institute of Technology, Vellore 632014, Tamil Nadu, India
* Correspondence: sowbiya.muneer@vit.ac.in or sobiyakhan126@gmail.com

**Abstract:** Salinity stress acts as a significant deterrent in the course of optimal plant growth and productivity, and mung bean, being a relay crop in the cereal cropping system, is severely affected by salinity. Silicon (Si), on the other hand, has exhibited promising outcomes with regards to alleviating salinity stress. In order to understand the critical mechanisms underlying mung bean (*Vigna radiata* L.) tolerance towards salt stress, this study examined the effects of different salinity concentrations on antioxidant capacity, proteome level alterations, and influence on Si-transporter and salt-responsive genes. Salinity stress was seen to effect the gaseous exchange machinery, decrease the soluble protein and phenolic content and NR activity, and increase the accumulation of reactive oxygen species. An efficient regulation of stomatal opening upon Si application hints towards proficient stomatal conductance and $CO_2$ fixation, resulting in efficient photosynthesis leading to proficient plant growth. The soluble protein and phenolic content showed improved levels upon Si supplementation, which indicates an optimal solute transport system from source to sink. The content of superoxide radicals showed a surge under salinity stress treatment, but efficient scavenging of superoxide radicles was noted under Si supplementation. Salinity stress exhibited more damaging effects on root NR activity, which was notably enhanced upon Si supplementation. Moreover, the beneficial role of Si was further substantiated as there was notable Si accumulation in the leaves and roots of salinity-stressed mung bean plants. Furthermore, Si stimulated competent ROS scavenging by reinforcing the antioxidant enzyme activity, as well coordinating with their isozyme activity, as expressed by the varying band intensities. Similarly, the Si-mediated increase in peroxidase activity may reveal changes in the mechanical characteristics of the cell wall, which are in turn associated with salinity stress adaptation. Proteomic investigations revealed the upregulation or downregulation of several proteins, which were thereafter identified by LC−MS/MS. About 45 proteins were identified and were functionally classified into photosynthesis (24%), metabolic process (19%), redox homeostasis (12%), transmembrane transport (10%), stress response (7%), and transcription regulation (4%). The gene expression analysis of the silicon transporter genes (*Lsi1*, *Lsi2*, and *Lsi3*) and SOS pathway genes (*SOS1*, *SOS2*, and *SOS3*) indicated the role of silicon in mitigating salinity stress. Hence, the findings of this study can facilitate a profound understanding of the potential mechanisms adopted by mung bean due to exogenous Si application during salinity stress.

**Keywords:** antioxidant activity; mung bean; mass spectrometer; proteomics; silicon; salinity stress

## 1. Introduction

Severe and persistent droughts in various regions of the world have compelled farmers to resort to low-quality irrigation water sources and to exploit unsustainable irrigation and fertilization approaches, which have escalated the problem of soil salinization [1]. Salinity stress negatively affects plant growth, development, and productivity as it reduces the osmotic potential of the rhizosphere, damages cell membranes, causes ionic imbalances

and photosynthesis impedance, and creates a significant increase in light-dependent respiration [2]. The interplay of ionic and osmotic stress coupled with nutrient deficiencies paves the way for oxidative stress development [3]. Because of their chemical makeup, ROS are extremely unstable and reactive and can start radical chain reactions that can deactivate proteins, oxidize membrane lipids, and harm nucleic acids [4]. Thus, unraveling the responses of plants towards salinity stress, in order to augment plant production, has been a pressing objective among plant breeders. In this regard, a comprehensive study on the mechanism of the molecular and biochemical responses of plants towards salinity stress tolerance is obligatory.

Mung bean (*Vigna radiata* L.), a leguminous crop, serves as a significant source of protein, carbohydrates, isoflavones, vitamins, fiber, and minerals [5]. However, salinity stress deleteriously disturbs plant germination, growth, the reproductive stage, and the capacity to biologically fix nitrogen in legumes. In mung bean, salinity has been observed to affect seedling germination and development [6], photosynthesis, nodulation [7], the accumulation of ROS, water status, membrane stability, and the content of pigments [7,8]. Of the various salinity management strategies, we intended to take advantage of the application of exogenous Si to combat salinity stress in mung bean because Si research so far has largely neglected legumes.

The position of silicon (Si) in terms of its "essentiality" for plant growth and development has been a reasonably debated topic among researchers. However, the plethora of research findings that have established Si as a proficient player in the alleviation of various abiotic stresses, such as salinity stress, drought stress, and metal toxicity, cannot be undermined either [7,9–11]. The protective role of Si is usually seen in plants due to the polymerization of silicates in the endodermis and exodermis. This leads to the obstruction of the $Na^+$ bypass route, resulting in lignification and suberization, as well as the formation of casparian bands, which disrupt the flow of solutes from roots to shoots by altering the properties of the membrane transport system [12]. Owing to the existence of precise transporters found in the cellular membranes of plant roots, silicon can be rapidly transported [13]. The soil-to-root influx of Si is mediated by the influx transporter *Lsi1* and its homolog *Lsi6*, whereas the efflux transporter *Lsi2* dictates the apoplastic release of Si, which is followed by Si-translocation to shoots mediated by a transpiration stream. Moreover, Si transporters *Lsi6* and *Lsi3* are involved in xylem unloading and re-loading, respectively [14].

Previously, efforts have been made to study the alterations in the protein expression of plants under salinity stress using proteomic approaches in alfalfa [15], maize [16], Halophytes *Suaeda maritima* (L.), and *Salicornia brachiate* [17]. However, the fact that very limited research has attempted to elucidate the effect of Si on the protein expression profiles of plants under salinity stress is concerning. Nevertheless, the proteomics analysis of tomato [18], capsicum [19], and rose [20] under salinity stress revealed a downregulation of the functional proteins, which were upregulated under Si supplementation. However, to the best of our knowledge, a proteomic analysis to elucidate Si's role in providing salinity stress resistance to mung bean is not available.

The orchestration of the multifaceted molecular events regulating the initiation or suppression of various salt-stress responsive genes, such as the salt overly sensitive (*SOS*) gene, culminate in conferring salt tolerance to plants [21]. Genes encoding the SOS proteins, which play a key role in maintaining a well-adjusted ion level inside the cell and providing salt tolerance, have been found in wheat [22], barley [23], and mustard [24]. Furthermore, plants overexpressing *SOS* genes have shown an increase in salt tolerance [25]. However, the role of Si in the regulation of *SOS* genes has not been widely studied as far as legumes are concerned.

It is of paramount importance to scrutinize and comprehend the underlying molecular mechanisms that are involved in the course of salt-stress tolerance in mung bean with Si supplementation, such that the improvement of this important legume crop is conceivable. To this point, proteome and transcriptome level analysis of mung bean under salinity

stress and Si supplementation has not been reported, which makes our study all the more indispensable and worth exploring. In this study, we aim to understand the mechanism behind protection against excessive ROS, connect the various dots related to changes in protein expression under salinity stress and Si supplementation using LC−MS/MS, and examine the role of mRNA level regulation of Si-transporter genes and salt-responsive genes in mung bean under salinity stress and Si supplementation. Investigation of mung bean protein expression patterns in response to salt stress will open up new avenues for understanding the regulatory networks of mung bean salt-stress acclimation and aid in the selection of candidate proteins for modification to increase salt-stress tolerance.

## 2. Materials and Methods

### 2.1. Plant Materials and Growth Conditions

This research was carried out at the polyhouse of School of Agricultural Innovations and Advanced Learning (VAIAL), Vellore Institute of Technology, Vellore, India. The polyhouse's growing conditions were as follows: temperature regime of 30 °C and 25 °C day and night, lighting period of 16:8 h, and relative humidity of approximately $65 \pm 5\%$. Mung bean (*Vigna radiata* L.) seeds were subjected to surface sterilization with 5% (*v/v*) sodium hypochlorite solution for 30 min before being rinsed in distilled water. Three seeds were sown in plastic pots (with a diameter of 13 cm and height of 17 cm). The pots were filled at a 1:1 ratio with sterilized red soil and vermicompost, and the soil salinity and pH were 0.35 dS m$^{-1}$ and 7.86, respectively.

### 2.2. Experimental Design

The experiment utilized a completely randomized design (CRD) with four replicates for each treatment. The plants were allowed to grow until they reached the vegetative stage (30 days after germination) and were then divided into eight groups for a combination of silicon (Si) (5 mM) and salinity treatments (10 mM NaCl, 20 mM NaCl, and 50 mM NaCl) for 10 days. The various treatments included the following: (a) control (T1) (b) –NaCl + Si (T2), (c) 10 mM NaCl/-Si (T3), (d) 10 mM NaCl/+Si (T4), (e) 20 mM NaCl/-Si (T5), (f) 20 mM NaCl/+Si (T6), (g) 50 mM NaCl/-Si (T7), and (h) 50 mM NaCl/+Si (T8). For silicon treatment, the plants were irrigated with a sodium silicate (5 mM) solution and intermittently irrigated with water for 10 days. The leaf and root samples were collected after 10 days of treatment and stored at −80 °C until further use.

### 2.3. Scanning Electron Microscope (SEM Analysis)

Fresh leaves were cut into small pieces, treated in 2.5% glutaraldehyde solution (pH7.4), stored at 4 °C, and then dehydrated using a wide range of ethanol concentrations (95–50%) for the SEM analysis. The leaves were further oven dried at 60 °C for 48 h [26]. A scanning electron microscope (model: EVO-18 Research, Carl Zeiss, Birmingham, UK) was used to examine the structure of the stomata.

### 2.4. Determination of Total Soluble Protein, Total Soluble Sugars and Total Phenolic Content

To determine the total soluble protein content, leaf samples (0.5 g) were homogenized in a mortar and pestle with 200 mM phosphate buffer (pH7), followed by centrifugation at 8000 rpm for 10 min. To 0.5 mL of supernatant, 10% TCA was added, followed by centrifugation at 3300 rpm for 30 min. The supernatant obtained was discarded, the pellets were then washed with water and dissolved in 1 mL of 0.1 N NaOH. Furthermore, 0.2 mL of the supernatant was mixed with 5 mL of Bradford reagent, then incubated for 5 min and the absorbance was read at 595 nm [27]. To determine the total soluble sugar content, the leaf samples (0.5 g) were homogenized in 5 mL of 80% ethanol, followed by centrifugation at 6000 rpm for 15 min. To the supernatant, 12.5 mL of 80% ethanol and 1 mL of 0.2% anthrone solution was added. The reaction was placed in a water bath at 100 °C for 10 min. The absorbance was read at 620 nm [28]. To determine the total phenolics content, 0.1 g of leaf samples were suspended in a test tube containing 1.5 mL of 50% (*v/v*) methanol and

1% (*v/v*) HCL. The reaction was placed in a water bath at 80 °C for 15 min. Furthermore, 0.02 mL of leaf extract (diluted in 0.08 mL extraction solution) was mixed with 0.7 mL of Folin−Ciocalteu solution (diluted in 1:10 ratio) and 0.7 mL of 6% (*w/v*) Na$_2$CO$_3$. The samples were placed in the dark for 1 h and the absorbance was measured at 765 nm [29]. Gallic acid was used as a standard.

### 2.5. Determination of Nitrate Reductase (NR) Activity

The NR activity was measured according to Lopez-serrano et al. [30]. Briefly, 0.2 g of leaf and root samples were immersed in 100 mM potassium phosphate buffer (pH 7.5), 1% (*v/v*) propanol, and 100 mM KNO$_3$, and then incubated in a hot water bath for 60 min at 30 °C. The reaction was stopped by placing the test tubes in a boiling water bath for 5 min. From this, 1 mL of supernatant was taken, to which 1 mL of 0.02% N-naphthyl ethylenediamine and 1 mL of 1% sulfanilamide were added and the absorbance was measured at 540 nm. To determine the quantity of NO$_2$ in the samples, a standard curve using KNO$_2$ was prepared.

### 2.6. Estimation of Silicon Concentration

From each pot, approximately 10 expanded leaves and masses of roots were selected at random. Each treatment group had five replications. The leaves were dried in a hot air oven at 80 °C for 48 h and then ground into a fine powder. Approximately 100 mg of leaf and root samples were acidified with HNO$_3$ for 12 h and further digested using the microwave-digestion method. Silicon concentrations were measured with a Perkin Elmer Optimum 5300 DV inductively coupled plasma optical emission spectrometer (ICP-OES) [9].

### 2.7. Determination of Hydrogen Peroxide (H$_2$O$_2$) and Superoxide (O$_2^-$) Content

The content of H$_2$O$_2$ was measured according to Velikova et al. [31] with minor modifications. Fresh leaves (0.25 g) were homogenized in 2 mL of 0.1% (*w/v*) trichloroacetic acid (TCA), followed by centrifugation at 10,000 rpm for 8 min at 4 °C. To the supernatant, 0.6 mL of 0.1% (*w/v*) TCA was added and it was then incubated for 1 h at room temperature in a dark place. Absorbance was read at 390 nm. The H$_2$O$_2$ content was calculated from a H$_2$O$_2$ standard curve. The O$_2^-$ content was measured according to Muneer et al. [9], with slight modifications.

### 2.8. Estimation of Antioxidants Enzyme Activity and Their Relative Staining

To measure the antioxidant enzyme activity, the leaf samples (0.1 g) were homogenized in an extraction buffer comprising 50 mM potassium phosphate buffer (pH7.0) with 1 mM EDTA, 0.05% triton X, and 1 mM polyvinylpyrrolidone (PVP); centrifugation was then carried out at 10,000 rpm for 20 min at 4 °C. The supernatant was then utilized to determine the antioxidant enzyme activity. The superoxide dismutase (SOD) activity was analyzed using the nitro blue tetrazolium (NBT) inhibition method of Giannopolitis and Ries et al. [32]. One unit (U) of SOD activity was defined as the quantity of the enzyme that inhibited the photochemical reduction of NBT by 50%. Catalase (CAT) activity was performed according to Manivannan et al. [19]. One unit of catalase decomposed 1.0 μmole of H$_2$O$_2$ per minute, while the H$_2$O$_2$ concentration declined from 10.3 mM to 9.2 mM. The APX activity was determined according to Nakano and Asada et al. [33]. One unit (U) of APX activity corresponded to the amount of enzyme required to oxidize 1 μmole of ascorbic acid per minute per mg of protein.

For native staining, the antioxidant enzymes (30 μg) were electrophoresed in 10% resolving and 4% stacking gel, respectively, for APX and CAT isozymes, whereas, 15% resolving gel and 5% stacking gel were used for separating the SOD isozymes at 4 °C for 4 h at 80 V in a Tris-Glycine (pH8.3) running buffer. The active staining of isozymes of SOD, CAT, and APX were performed according to Pham et al. [34].

### 2.9. Native PAGE Profiling of Isozymes of Peroxidases' Enzyme(s)

The leaf samples (0.5 g) were homogenized in an extraction buffer composed of 100 mM K-PO$_4$ buffer (pH7.0) and 2 mM phenylmethylsulphonyl fluoride, and centrifugation was done at 14,000 rpm for 20 min at 4 °C. Active staining of GPOX, SPOX, and BPOX was carried out according to Lee et al. [35].

### 2.10. Protein Extraction and One-Dimensional Gel Electrophoresis (SDS-PAGE)

Protein extraction and SDS-PAGE were performed according to Muneer et al. [36]. The protein content was measured using the Bradford test and a standard curve of bovine serum albumin (BSA). After electrophoresis, the gel was stained with Comassie brilliant blue stain (CBBS), which is commercially available (Bio-Rad).

### 2.11. In-Gel Digestion of Protein Bands and Mass Spectrometer Analysis

The method of Muneer et al. [37] was used for the protein in gel digestion (for a detailed methodology of the in-gel digestion, please refer to Muneer et al.) [37].

The MS and MS/MS spectra data were analyzed with a mass tolerance of 50 ppm using the NCBI and Protein Pilot V.3.0 database software (with the MASCOT V.2.3.02 database search engine). Oxidation of methionines and carbamidomethylation of cysteines were permitted for database searches of the MS/MS spectra. A statistically significant threshold value of $p = 0.05$ was used to search for individual peptide ion scores. According to the gene ontology analysis (http://www.geneontology.org, accessed on 1 August 2022), the identified proteins were further categorized on the basis of the biological processes in which they contribute. The identified proteins were also analyzed to observe possible protein−protein interactions using the STRING database.

### 2.12. RNA Isolation, cDNA Preparation, and RT-PCR

RNA was isolated from the leaves using an RNA isolation Kkit according to the manufacturer's instructions (Hi-Media). Real-time PCR was carried out in Applied Biosystems using SYBR Green Chemistry (Sensifast HiRoxkit Bioline, Memphis, TN, USA) for 5 min at 95 °C, followed by 35 cycles of 20 s at 95 °C, 30 s at 57 °C, and 30 s at 72 °C, followed by 10 min at 72 °C. Actin was utilized to normalize all quantifications. For the RT-PCR reactions and qPCR, three distinct RNA preparations from independently grown plants were utilized. The results were analyzed using qBase plus 13 software. Table 1 lists the gene-specific primers utilized in our investigation.

**Table 1.** Primer sequences used for the RT-PCR analysis.

| Gene | Forward Sequence (5′---------3′) | Reverse Sequence (5′---------3′) |
|---|---|---|
| *Lsi-1* | ATGGAGAGTGAAGGAGGGAA | TTAGAGGGTAACACATTGTT |
| *Lsi-2* | CGATGACTTTGCCCATCGTG | GCAATATGAACCTCGTCCGC |
| *Lsi-3* | TATTTYTTCCTGGCCAACCT | TTAAGCTATAGATGAGGGGG |
| *SOS1* | GCCAGCTATAAGCTAAGCAC | GCAATCCCTAAAGCAAGACC |
| *SOS2* | GCATTCATCGTGCAGCATC | GTATAGTCTCGCCATCACCTC |
| *SOS3* | ACGAAGAATTTCAGCTCGC | TCACCTAACTCGATGACTCC |
| *Actin* | ATCCTCCGTCTTGACCTTG | TGTCCGTCAGGCAACTCAT |

### 2.13. Statistical Analysis

For the physiological parameters, a complete randomized design was employed with four replicates. The percentage change was calculated using: [(Treatment−Control)/ (Treatment) × 100]. To compare the means of distinct replicates, Tukey's studentized range test was applied. Unless otherwise noted, the results are based on differences between means, with a level of significance of $p < 0.05$.

## 3. Results

*3.1. Effect of Salinity Stress on Structure and Opening/Closing of Stomatal Pore of Mung Bean Supplemented with Si*

Salinity stress impairs photosynthetic machinery and thus adversely affects the gaseous exchange in plants subjected to abiotic stress conditions by intervening with the opening and closing of the stomata. Hence, in our study, after 10 days of salinity stress treatment, the stomatal structure was found to be affected (Figure 1). It was evident that the stomatal pore was found to be closed when different concentrations (T3, T5, and T7) of salinity stress were provided when compared with the control. On the contrary, the stomatal opening was observed when Si was supplemented to the salinity stress-treated plants in T4, T6, and T8, respectively.

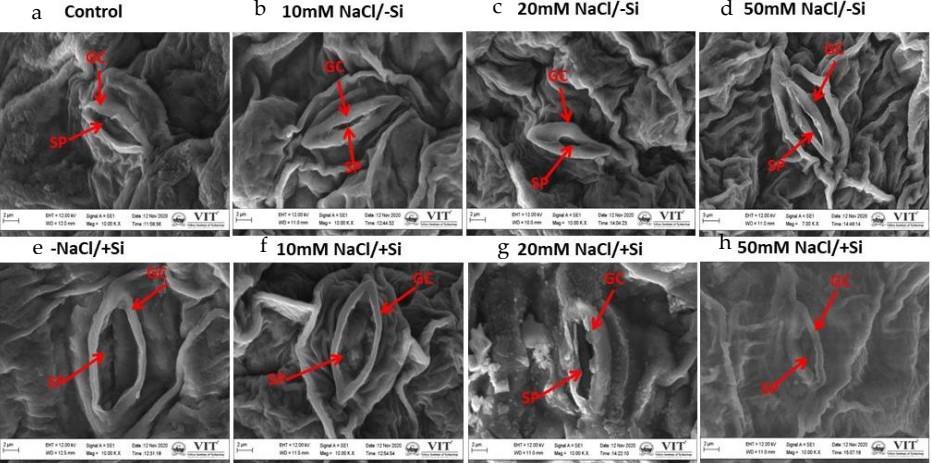

**Figure 1.** Representative image of stomatal opening/closing of mung bean (*Vigna radiata*) under Si supply (5 mM) and salinity stress (**a**) Control (T1) (**b**) –NaCl + Si (T2) (**c**) 10 mM NaCl/-Si (T3) (**d**) 10 mM NaCl/+Si (T4) (**e**) 20 mM NaCl/-Si (T5) (**f**) 20 mM NaCl/+Si (T6) (**g**) 50 mM NaCl/-Si (T7) (**h**) 50 mM NaCl/+Si (T8) for a duration of 10 days.

*3.2. Effect of Salinity Stress on the Soluble Protein, Sugar, and Phenolic Content of Mung Bean Supplemented with Si*

The content of soluble protein was seen to be reduced by 25% in the highest concentration (T7) of salinity stress provided when compared with the control (T1) (Figure 2A). However, Si supplementation increased the levels of soluble protein by 93% in T4 when compared with T3, but reduced the soluble protein content in T6 and did not have many significant changes in T8. The total soluble sugar content in salinity-treated plants was seen to be increased upon Si supplementation in T4, but there was no significant difference in T6 and T8 upon the supplementation of Si (Figure 2B). For the total phenolic content, the phenolics levels were found to be increased through the supplementation of Si to e-salinity-stressed plants (Figure 2C). The phenolic content was increased by 53% and 50% in T6 and T8, respectively, when compared with the salinity treatments. However, no significant change was observed upon Si supplementation in T4.

*3.3. Effect of Salinity Stress on the Nitrate Reductase Activity of Mung Bean Supplemented with Si*

Salinity stress was seen to affect the NR activity in the leaves in T3 when compared with the control, but it did not show significant changes in the root NR activity (Figure 3A,B). The NR activity in the leaves was seen to be reduced by 38% in T3 when compared with the control (T1). After supplementation with Si, the NR activity in leaves did not show significant changes. However, after Si supplementation, the root NR activity was found to be increased, and the most significant increase was that of 92% in T4 and 59% in T8. However, no significant change was observed in T6 upon Si supplementation.

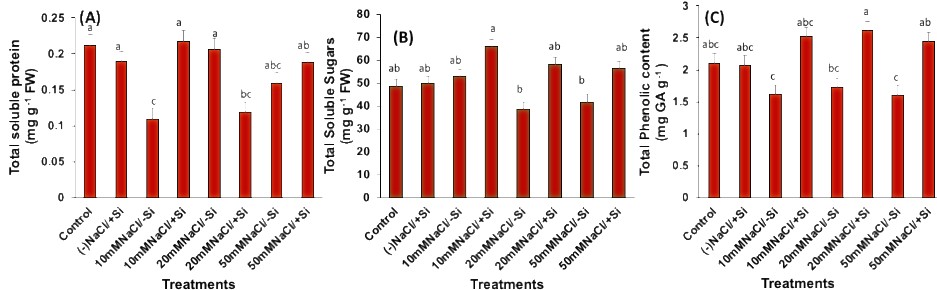

**Figure 2.** Changes in content of the (**A**) total soluble protein, (**B**) total soluble sugars, and (**C**) total phenolic contents of mung bean (*Vigna radiata*) under Si supply (5 mM) and salinity stress: (a) Control (T1), (b) –NaCl + Si (T2), (c) 10 mM NaCl/-Si (T3), (d) 10 mM NaCl/+Si (T4), (e) 20 mM NaCl/-Si (T5), (f) 20 mM NaCl/+Si (T6), (g) 50 mM NaCl/-Si (T7), and (h) 50 mM NaCl/+Si (T8) for a duration of 10 days. Vertical bars indicate mean $\pm$ SE of the mean for $n = 4$. Means denoted by different letter are significantly different at $p \leq 0.05$ according to Tukey's studentized range test.

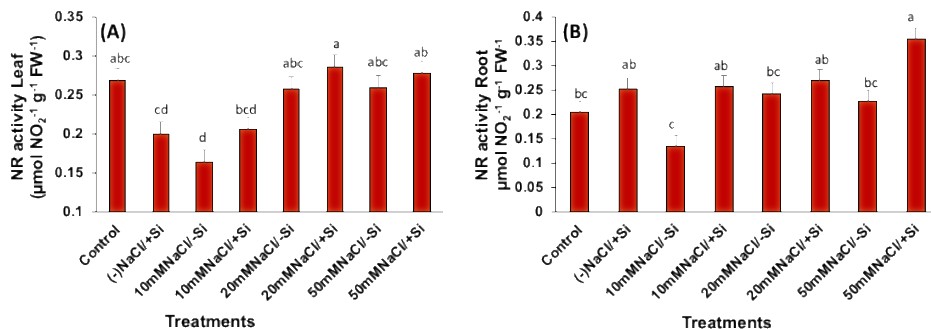

**Figure 3.** Changes in NR activity for the (**A**) leaves and (**B**) roots of mung bean (*Vigna radiata*) under Si supply (5 mM) and salinity stress: (a) control (T1), (b) –NaCl + Si (T2), (c) 10 mM NaCl/-Si (T3), (d) 10 mM NaCl/+Si (T4), (e) 20 mM NaCl/-Si (T5), (f) 20 mM NaCl/+Si (T6), (g) 50 mM NaCl/-Si (T7), and (h) 50 mM NaCl/+Si (T8) for a duration of 10 days. Vertical bars indicate mean $\pm$ SE of the mean for $n = 4$. Means denoted by a different letter are significantly different at $p \leq 0.05$ according to the Tukey's studentized range test.

### 3.4. Silicon Concentration in Leaves and Roots

The Si content in the leaves was slightly more than the Si content in the roots. In the leaves, the highest Si content was observed in T2 and T8 (Figure 4A). The Si content increased in T4, T6, and T8 by 21%, 49%, and 81%, respectively. The highest change in Si content was observed in T8 compared with T7. In the roots, the Si content was the highest in T4 followed by T6 and T8 (Figure 4B). The root Si content in T6 and T8 did not show a significant difference when compared with the Si alone treatment (T2). The Si content in T3 and T5 did not show significant changes either. However, the highest significant percentage change of 51% was observed in T4 when compared with T3.

### 3.5. Effects of Salinity Stress on Hydrogen Peroxide ($H_2O_2$) and Superoxide ($O_2^-$) Content of Mung Bean Supplemented with Si

Following salinity stress treatments for 10 days, the $H_2O_2$ content was found to have increased by 73.2% in T3 compared with the control (Figure 5A). However, the $H_2O_2$ content was found to be decreased in T5 and T7, but only the decrease in T7 was significant. Moreover, after the supplementation of Si, the content of $H_2O_2$ did not significantly reduce in T4 and T8, but a significant reduction of 49% was seen in T6 when compared with T5. For the $O_2^-$ content, the levels increased in salinity treatments T3, T5, and T7 by 238%, 270%, and 217%, respectively, when compared with the control (T1) (Figure 5B). After Si supplementation in T4 and T6, the levels of $O_2^-$ were found to be reduced significantly by

32% and 31%, respectively, but the levels of $O_2^-$ increased in T8 when compared with T7 after Si supplementation.

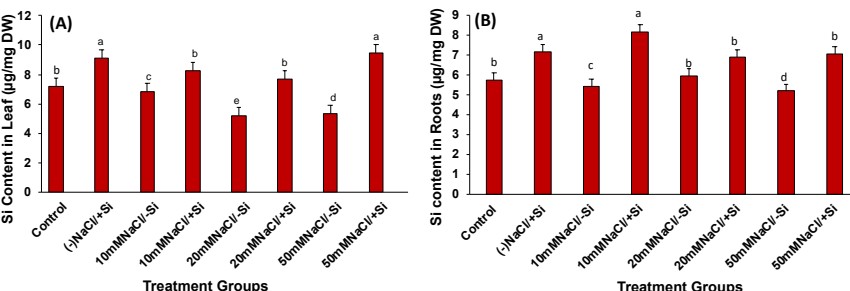

**Figure 4.** Changes in Si content of (**A**) leaves and (**B**) roots of mung bean (*Vigna radiata*) under Si supply (5 mM) and salinity stress: (a) control (T1), (b) –NaCl+Si (T2), (c) 10 mM NaCl/-Si (T3), (d) 10 mM NaCl/+Si (T4), (e) 20 mM NaCl/-Si (T5), (f) 20 mM NaCl/+Si (T6), (g) 50 mM NaCl/-Si (T7), and (h) 50 mM NaCl/+Si (T8) for a duration of 10 days. Vertical bars indicate mean ± SE of the mean for *n* = 4. Means denoted by s different letter are significantly different at *p* ≤ 0.05 according to the Tukey's studentized range test.

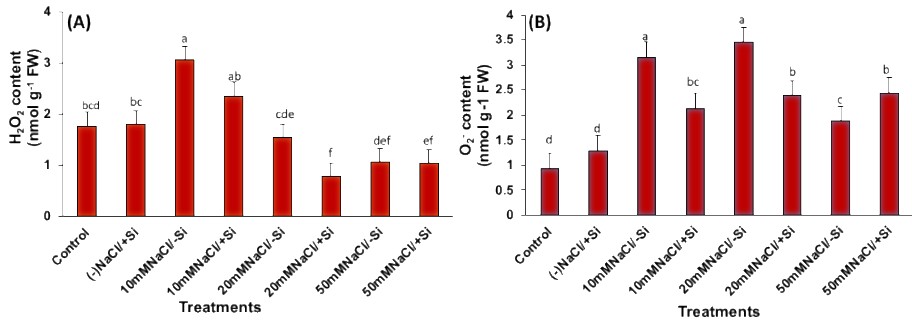

**Figure 5.** Changes in the content of (**A**) $H_2O_2$ and (**B**) $O_2^-$ of mung bean (*Vigna radiata*) under Si supply (5 mM) and salinity stress: (a) control (T1), (b) –NaCl + Si (T2), (c) 10 mM NaCl/-Si (T3), (d) 10 mM NaCl/+Si (T4), (e) 20 mM NaCl/-Si (T5), (f) 20 mM NaCl/+Si (T6), (g) 50 mM NaCl/-Si (T7), and (h) 50 mM NaCl/+Si (T8) for a duration of 10 days. Vertical bars indicate mean ± SE of the mean for *n* = 4. Means denoted by a different letter are significantly different at *p* ≤ 0.05 according to the Tukey's studentized range test.

*3.6. Effect of Salinity Stress on Antioxidant Activity and Their Isozyme Patterns in Mung Bean Supplemented with Si*

The SOD activity was seen to be affected in all of the salinity treatment groups, with the exception of T3 (Figure 6A). Si supplementation was, however, seen to enhance the activity of SOD significantly by 185% and 101% in T6 and T8, respectively, when compared with T5 and T7, respectively. However, a significant difference between SOD activity in the Si-treatment alone and control was missing. Moreover, Si supplementation in T4 was found to have decreased the SOD activity when compared with T3. The isozyme bands of SOD were found to be more intense in the treatment groups where Si was supplemented along with salinity stress (Figure 6D). For the isozymes of SOD, SOD-3 displayed a reduced band intensity in T3, T5, and T6; however, after Si supplementation, SOD-3 had greater band intensities in T4, T6, and T8, respectively. The SOD-2 isozyme bands were also expressed more in T4 compared with T3. The APX activity increased by 120% and 48% in T6 and T8, respectively, when compared with T5 and T7, respectively, upon Si supplementation (Figure 6B). However, there was no significant difference between the APX activity in the Si-treatment alone and the control. Moreover, Si supplementation in T4 was found to have decreased the APX activity when compared with T3. The band intensity of isozyme APX-2 diminished under salinity treatments T3 and T5, whereas Si supplementation in T4 and T6

showed increased band intensities of APX-2 (Figure 6D). The CAT activity followed the same trend as observed in the SOD and APX activity. After Si supplementation, the activity of CAT increased by 241% and 26% in T6 and T8, respectively, when compared with T5 and T7, respectively (Figure 6C). However, the CAT for T3 and T4 were not significant when compared with T2 and T1, respectively. Moreover, Si supplementation in T4 was found to have decreased the CAT activity when compared with T3. Of the two CAT isozymes stained, CAT-2 showed expression changes across salinity stress treatments and Si supplemented groups (Figure 6D). CAT-2 isozyme bands showed lesser band intensity in T3 and were highly expressed in T3 and T5, whereas the expression of CAT-2 was increased in T4 and T6, respectively.

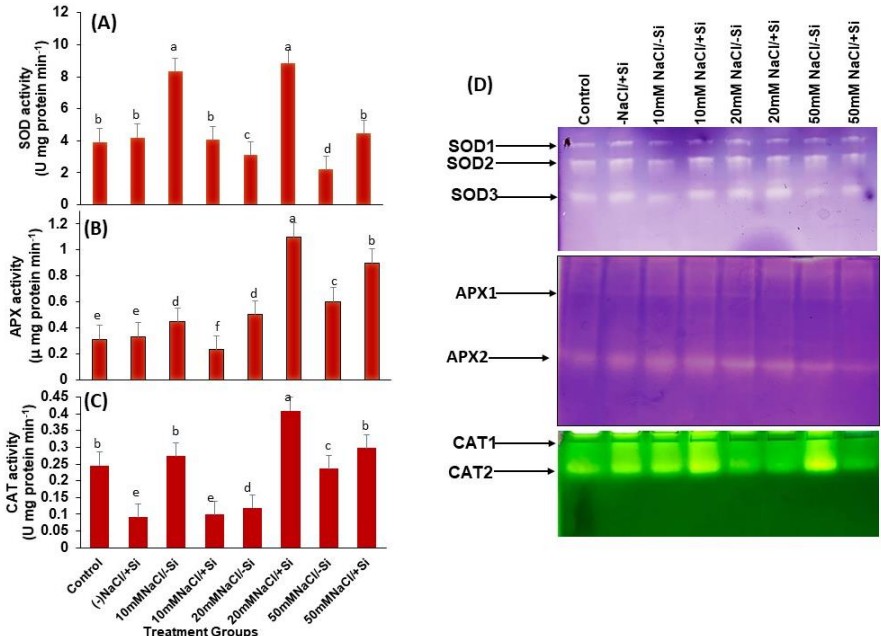

**Figure 6.** Changes in the antioxidant enzyme activity and isozyme profiles of (**A**,**D**) superoxide dismutase, (**B**,**D**) ascorbate peroxidase, and (**C**,**D**) catalase of mung bean (*Vigna radiata*) under Si supply (5 mM) and salinity stress: (a) control (T1), (b) –NaCl+Si (T2), (c) 10 mM NaCl/-Si (T3), (d) 10 mM NaCl/+Si (T4), (e) 20 mM NaCl/-Si (T5), (f) 20 mM NaCl/+Si (T6), (g) 50 mM NaCl/-Si (T7), and (h) 50 mM NaCl/+Si (T8) for a duration of 10 days. Vertical bars indicate mean ± SE of the mean for *n* = 4. Means denoted by a different letter are significantly different at $p \leq 0.05$ according to Tukey's studentized range test.

*3.7. Effect of Salinity Stress on the Isozymes of Peroxidase Enzymes' in Mung Bean Supplemented with Si*

The active staining of BPOX revealed four isozymes, BPOX-1, BPOX-2, BPOX-3, and BPOX-4, among which the expression pattern on BPOX-4 was seen to be more prominent (Figure 7A). BPOX-4 had a lower band intensity in T5 and T7; however, after Si supplementation in T6 and T8, the band intensities were seen to have increased. Similarly, the GPOX isozyme GPOX-2 was seen to have a higher expression profile under Si supplementation in T6 and T8 when compared with T5 and T7, respectively (Figure 7B). SPOX isozymes did not display many changes in expression among the salinity-treated groups and Si-supplemented groups (Figure 7C). However, under Si supplementation, the band intensity of SPOX-2 was found to be higher in T8 compared with that of T7.

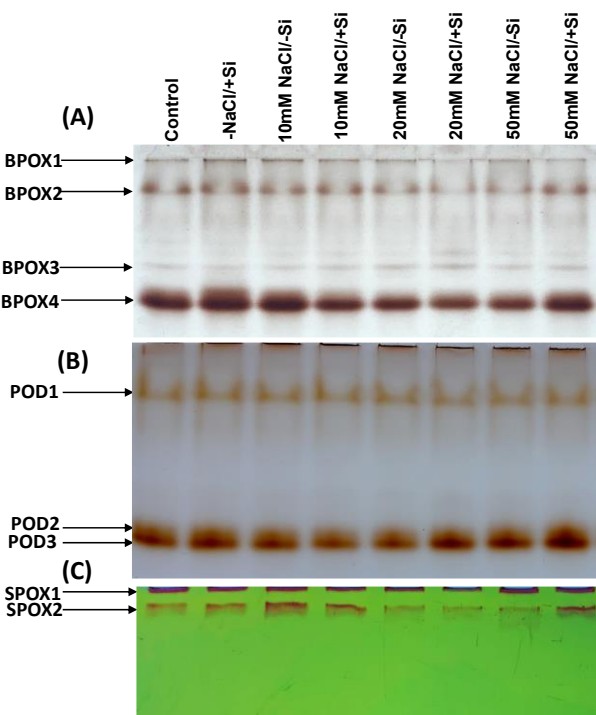

**Figure 7.** Profiles of peroxidase isozymes: (**A**) BPOX, (**B**) POD, and (**C**) SPOX of mung bean (*Vigna radiata*) under Si supply (5 mM) and salinity stress: (a) control (T1), (b) –NaCl+Si (T2), (c) 10 mM NaCl/-Si (T3), (d) 10 mM NaCl/+Si (T4), (e) 20 mM NaCl/-Si (T5), (f) 20 mM NaCl/+Si (T6), (g) 50 mM NaCl/-Si (T7), and (h) 50 mM NaCl/+Si (T8) for a duration of 10 days.

### 3.8. Changes in the Expression of Proteins

The protein profile was analyzed by one-dimensional gel electrophoresis (SDS-PAGE), and the proteins were observed to be either upregulated or downregulated (Figure 8, graphical representation shown in Figure 9B). Following this, these up or downregulated proteins were identified using a mass spectrometer (LC−MS/MS) (Table 2). Direct gene ontology consortium (http://www.geneontology.org/, accessed on 1 August 2022) was used to determine the percent variation of identified proteins for functional categorization. All of the differentially expressed proteins were grouped into photosynthesis (24%), metabolic process (19%), redox homeostasis (12%), transmembrane transport (10%), stress response (7%), and transcription regulation (4%) (Figure 9A). Additionally, the STRING database was used to examine the identified proteins for any protein−protein interactions (Figure 10). The proteins mostly interacted with cell division, ATP synthase, photosynthesis, transport, metabolism, and other signaling-pathway-related proteins.

#### 3.8.1. Proteins Related to Photosynthesis

Salinity stress was found to have decreased the expression of vital proteins that are involved in photosynthesis, such as ribulose bisphosphate carboxylase large chain (band 3A, 3C, 7A, 7B, and 7C). However, Si supplementation positively influenced the expression of ribulose bisphosphate carboxylase large chain (band 6A and 6D), thus restoring the normal functioning of the photosynthetic process.

#### 3.8.2. Proteins Related to Metabolic Processes

Salinity stress was seen to downregulate the enzyme fumarylacetoacetase (band 3B) involved in tyrosine and phenylalanine catabolism, fructose-bisphosphatase (band 3G) and putative phosphoketolase (band 3G) involved in carbohydrate metabolism, and putative phospho-2-dehydro-3-deoxyheptonate aldolase (band 3H) involved in amino acid synthesis. After Si supplementation to the salinity-stressed plants, enzyme 1-phosphatidylinositol 4-kinase (band 7D) involved in lipid metabolism, fructose-bisphosphate aldolase (band

7F) involved in carbohydrate metabolism, cyclase family protein (7F) involved in amino acid metabolism, tRNA(Ile)-lysidine synthetase (band 7F) involved in tRNA metabolic process, and precorrin-2 dehydrogenase (band 7H) involved in porphyrin biosynthesis were observed to be upregulated. Moreover, Si supplementation alone could upregulate various key proteins such as cellulose (band 2C), peptide hydrolase (band 2F), 3-deoxy-7-phosphoheptulonate synthase (band 2G), and UDP-N-acetylglucosamine transferase subunit ALG13 (band 2I) involved in key metabolic processes.

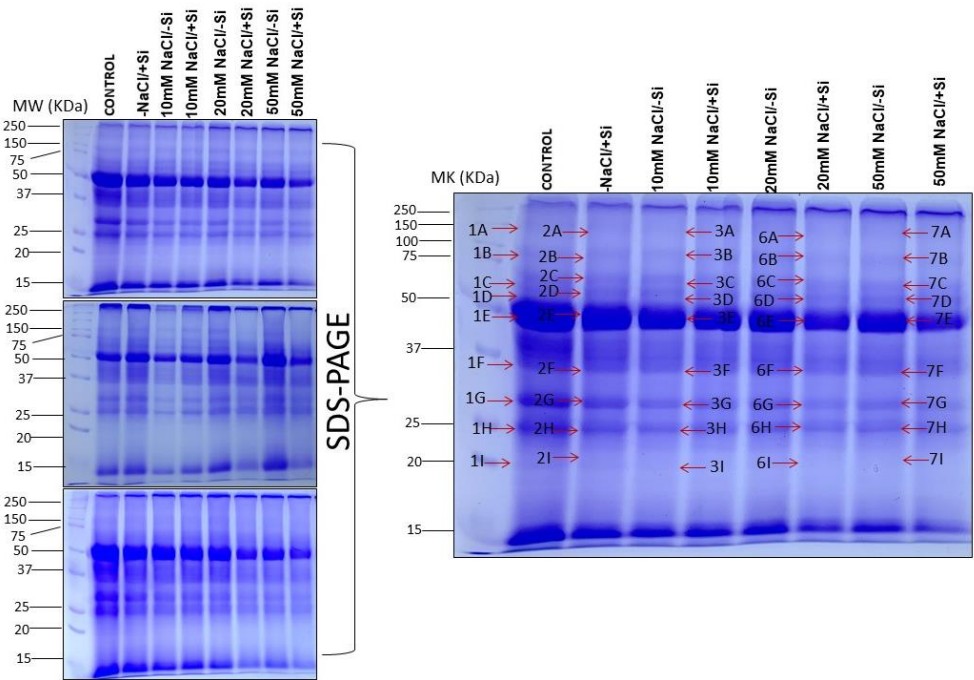

**Figure 8.** Representative image of protein profiles (SDS-PAGE) of mung bean (*Vigna radiata*) under Si supply (5 mM) and salinity stress: (a) control (T1), (b) –NaCl+Si (T2), (c) 10 mM NaCl/-Si (T3), (d) 10 mM NaCl/+Si (T4), (e) 20 mM NaCl/-Si (T5), (f) 20 mM NaCl/+Si (T6), (g) 50 mM NaCl/-Si (T7), and (h) 50 mM NaCl/+Si (T8) for a duration of 10 days. Differentially expressed bands excised for protein identification by LC−MS/MS are marked by arrows.

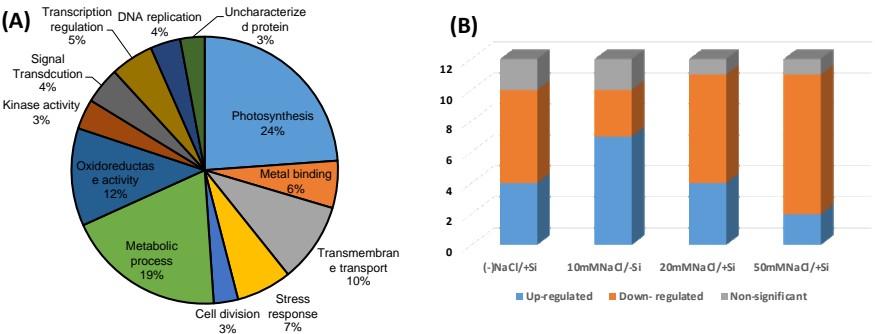

**Figure 9.** Comparative analysis of the proteome profiles between the treatments. (**A**) Functional classification of the proteins identified by Gene ontology analysis and (**B**) Venn diagram illustration of the up, down, or non-significantly regulated proteins.

**Table 2.** Identification of differentially-expressed proteins by LC/MS-MS in mung bean (*Vigna radiata*).

| Band no. | Protein Name | Plant Species | Accession Number | Protein Score | Biological Function |
|---|---|---|---|---|---|
| **1A** | Fumarylacetoacetase | *Cephalotus follicularis* | A0A1Q3BBE1 | 28 | Metal ion binding, chaperone binding |
| | Uncharacterized protein | *Apolygus lucorum* | A0A6A4KDN9 | 26 | Integral component of membrane |
| | Protein kinase domain-containing protein | *Rhizophagus irregularis* | U9V622 | 26 | ATP binding, protein kinase activity |
| **1B** | Ribulose bisphosphate carboxylase large chain | *Alfaroa guanacastensis* | A0A068L6A4 | 43 | Photorespiration, photosynthesis |
| | 5B protein like protein | *Arabidopsis thaliana* | Q9SUZ2 | 30 | Stress response |
| | DHA1 family multidrug resistance protein-like MFS transporter | *Paenibacillus prosopidis* | A0A368W0U8 | 29 | Transmembrane transport |
| **1C** | Ribulose bisphosphate carboxylase large chain | *Soleirolia soleirolii* | A0A0F7C9I4 | 101 | Photosynthesis |
| | Uracil permease | *Paludibacterium purpuratum* | A0A4R7BCH9 | 39 | Transmembrane transport |
| | Putative metallothionein expression activator | *Diaporthe ampelina* | A0A0G2HMQ0 | 35 | Metal binding |
| **1D** | Ribulose bisphosphate carboxylase large chain | *Trifolium repens* | A0A023HPA0 | 186 | Photosynthesis |
| | Peptidyl-tRNA hydrolase | *Glycomyces artemisiae* | A0A2T0UIK1 | 45 | Translation |
| | Flavodoxin | *Eggerthella lenta* | A0A369MMS0 | 42 | Metal binding |
| **1E** | Ribulose bisphosphate carboxylase large chain | *Berchemia lineata* | A0A7L8XJV8 | 223 | Photosynthesis |
| | Ribulose bisphosphate carboxylase large chain | *Pycnarrhena cauliflora* | B3FWZ0 | 223 | Photosynthesis |
| | Ribulose bisphosphate carboxylase large chain | *Soleirolia soleirolii* | A0A0F7C9I4 | 193 | Photosynthesis |
| **1F** | Biosynthetic peptidoglycan transglycosylase | *Xanthomonas arboricola* | A0A7W9QLL5 | 30 | Peptidoglycan synthesis |
| | LRRNT_2 domain-containing protein | *Quercus lobata* | A0A7N2MTJ2 | 30 | Transmembrane transport |
| | Cell division protein FtsQ | *Aeromonas veronii* | A0A6S4V1U8 | 29 | Cell division |
| **1G** | Ribulose bisphosphate carboxylase large chain | *Agathis borneensis* | Q9MVV3 | 61 | Photosynthesis |
| | Lysine-specific demethylase 3A | *Capsicum chinense* | A0A2G3CFW6 | 33 | Methylation |
| | Glucose-1-phosphate adenylyl transferase | *Rhodospirillaceae bacterium* | A0A2E5LIM1 | 31 | Glycogen biosynthetic process |
| **1H** | Lysozyme | *Enterobacter phage vB_EkoM5VN* | A0A7I8HQY3 | 32 | Defense response, catabolic process |
| | Putative N-glycosyltransferase | *Frankia alni* | Q0RGF9 | 32 | Transferase activity |
| | probable transcription factor KAN4 | *Juglans regia* | A0A2I4EQ27 | 31 | Transcription regulation |
| **1I** | Ribulose bisphosphate carboxylase small subunit | *Arachis duranensis* | A0A6P4D9J6 | 94 | Photosynthesis |
| | Uncharacterized protein | *Marchantia polymorpha* | A0A2R6WJ30 | 31 | NIL |
| | Uncharacterized protein | *Pseudocercospora fijiensis* | M2ZM51 | 31 | NIL |
| **2A** | Ribulose bisphosphate carboxylase large chain | *Trifolium repens* | A0A023HPA0 | 39 | Photosynthesis |
| | Uncharacterized protein | *Setaria italica* | K4A228 | 32 | NIL |
| | Endonuclease/exonuclease/phosphatase family metal-dependent hydrolase | *Rhizobium pisi* | A0A7W5BJJ3 | 27 | Endonuclease activity |

**Table 2.** *Cont.*

| Band no. | Protein Name | Plant Species | Accession Number | Protein Score | Biological Function |
|---|---|---|---|---|---|
| **2B** | tRNA wybutosine-synthesizing protein 4 | *Trichoderma arundinaceum* | A0A395NCK0 | 38 | Endonuclease activity |
| | ABC-type branched-subunit amino acid transport system substrate-binding protein | *Streptomyces* sp. BK022 | A0A4Q7Z6F6 | 34 | Transmembrane transport |
| | TPX2 domain-containing protein | *Marchantia polymorpha* subsp. *ruderalis* | A0A176W329 | 33 | Kinase activity, cell cycle/division |
| **2C** | Uncharacterized protein | *Kribbella* sp. VKM Ac-2527 | A0A4R6KBE5 | 35 | NIL |
| | Cellulase | *Bosea* sp. AK1 | A0A542B873 | 30 | Metabolic process |
| | Multiple sugar transport system permease protein | *Kribbella* sp. VKM Ac-2569 | A0A4Q7QE40 | 29 | Transmembrane transport |
| **2D** | Ribulose bisphosphate carboxylase large chain | *Kayea stylosa* | Q8MCX9 | 213 | Photosynthesis |
| | Precorrin-2 dehydrogenase | *Winogradskyella arenosi* | A0A368ZJY7 | 53 | Oxidoreductase |
| | Uncharacterized protein | *Klebsormidium nitens* | A0A1Y1HR37 | 47 | Transmembrane transport |
| | Haemolysin activation/secretion protein | *Cupriavidus plantarum* | A0A316F4A6 | 43 | Protein transport |
| **2E** | Ribulose bisphosphate carboxylase large chain | *Cornus eydeana* | Q2TV61 | 143 | Photosynthesis |
| | Ribulose bisphosphate carboxylase large chain | *Crossostylis grandiflora* | A0ZQX2 | 120 | Photosynthesis |
| | Acyltransferase | *Tardiphaga robiniae* | A0A7G6TUN4 | 41 | Transmembrane transport, transferase activity |
| **2F** | Uncharacterized protein | *Salix brachista* | A0A5N5L7N0 | 33 | Electron transport |
| | Peptide hydrolase | *Trichoderma arundinaceum* | A0A395NEC7 | 33 | Catabolic process |
| | Shugoshin_C domain-containing protein | *Kalanchoe fedtschenkoi* | A0A7N0U758 | 33 | Cell cycle/division |
| **2G** | Ribulose bisphosphate carboxylase large chain | *Aloe vera (Aloe)* | Q6VW13 | 82 | Photosynthesis/ photorespiration |
| | Nitronate monooxygenase | *Paraburkholderia unamae* | A0A328WWJ8 | 33 | Nitronate monooxygenase activity |
| | 3-deoxy-7-phosphoheptulonate synthase | *Paenibacillus peoriae* | A0A0K2F5Q8 | 30 | Metabolic process |
| **2H** | Superoxide dismutase | *Phaseolus lunatus* | Q3S614 | 46 | Stress response |
| | Histidine kinase | *Massilia aurea* | A0A7W9U5Y5 | 34 | Signaling, protein modification process |
| | Positive regulator of purine utilization | *Pyrenophora seminiperda CCB06* | A0A3M7MHM2 | 34 | Transcription, metal binding |
| **2I** | Ribulose bisphosphate carboxylase large chain | *Kalanchoe fedtschenkoi* | A0A7N0TKL3 | 141 | Photosynthesis/ photorespiration |
| | UDP-N-acetylglucosamine transferase subunit ALG13 | *Botryotinia fuckeliana* | A0A384JFM8 | 30 | Glycosylation, metabolic process |
| | 2-keto-3-deoxy-L-fuconate dehydrogenase | *Rhizobium* sp. PP-F2F-G48 | A0A4R1X109 | 28 | Oxidoreductase |
| **3A** | Ribulose bisphosphate carboxylase large chain | *Trifolium repens* | A0A023HPA0 | 88 | Photosynthesis/ photorespiration |
| | DUF4328 domain-containing protein | *Streptomyces violarus* | A0A7W4ZSR6 | 41 | Transmembrane transport |
| | PAS domain S-box-containing protein | *Mucilaginibacter* sp. E4BP6 | A0A7Y9HYI9 | 38 | Signaling, kinase activity |

**Table 2.** *Cont.*

| Band no. | Protein Name | Plant Species | Accession Number | Protein Score | Biological Function |
|---|---|---|---|---|---|
| **3B** | Fumarylacetoacetase | *Cephalotus follicularis* | A0A1Q3BBE1 | 26 | Catabolic process |
| | Protein kinase domain-containing protein | *Phaeosphaeria nodorum* | Q0UFQ8 | 26 | Transcription, phosphorylation |
| | Uncharacterized protein | *Prunus persica* | A0A251QEC7 | 25 | Defense response |
| **3C** | Ribulose bisphosphate carboxylase large chain | *Trifolium repens* | A0A023HPA0 | 96 | Photosynthesis/ photorespiration |
| | Ribulose bisphosphate carboxylase large chain | *Spirodela polyrhiza* | A0A0F7EWB6 | 96 | Photosynthesis/ photorespiration |
| | Ribulose bisphosphate carboxylase large chain | *Cladopus austro-osumiensis* | O03046 | 93 | Photosynthesis/ photorespiration |
| **3D** | Ribulose bisphosphate carboxylase large chain | *Vanilla planifolia* | A0A0D3M9U3 | 151 | Photosynthesis/ photorespiration |
| | Uncharacterized protein | *Lactuca sativa* | A0A2J6KPN6 | 33 | Protein auto phosphorylation |
| | ATPase subunit of ABC transporter with duplicated ATPase domains | *Rhizobium* sp. BK049 | A0A7W5KML0 | 31 | ATP binding |
| **3E** | Uncharacterized protein | *Chara braunii* | A0A388KU15 | 31 | Polymerase activity, DNA integration |
| | ATP-binding protein | *Raoultella ornithinolytica* | A0A225U1S1 | 29 | ATP binding |
| | Methyltransferase family protein | *Cellulomonas* sp. PhB143 | A0A3N2JFI2 | 29 | Methylation |
| **3F** | Chromosome partition protein Smc | *Cohnella lupini* | A0A3D9IW82 | 35 | DNA replication, chromosome condensation |
| | Long-chain acyl-CoA synthetase | *Streptomyces* sp. BK239 | A0A4Q7XQB8 | 34 | Aspartic-type endopeptidase activity |
| | SOS response UmuD protein | *Arthrobacter* sp. SLBN-122 | A0A542G4S1 | 32 | SOS response, DNA repair, transcription |
| | Diacylglycerol kinase iota | *Aegilops tauschii* | N1QUQ4 | 32 | Defense response |
| **3G** | Fructose-bisphosphatase | *Brassica napus (Rape)* | A0A078FJK5 | 45 | Metabolic process |
| | Putative phosphoketolase | *Fusarium culmorum* | A0A2T4H188 | 38 | Carbohydrate metabolic process |
| | Ferredoxin-NADP reductase | *Xanthomonas campestris* | A0A7W6KYF2 | 34 | Oxidoreductase |
| **3H** | Superoxide dismutase | *Glycine max* | Q71UA1 | 60 | Stress response |
| | GntR family transcriptional regulator | *Rathayibacter* sp. PhB93 | A0A3N1NHP1 | 37 | Transcription |
| | Putative phospho-2-dehydro-3-deoxyheptonate aldolase | *Phaeomoniella chlamydospora* | A0A0G2EP81 | 36 | Amino acid biosynthesis, metal binding |
| **3I** | Ribulose bisphosphate carboxylase small subunit | *Arachis duranensis* | A0A6P4D9J6 | 80 | Photosynthesis/ photorespiration |
| | Uncharacterized protein | *Punica granatum* | A0A218W2T9 | 31 | Transcription |
| | UDP-N-acetylglucosamine transferase subunit ALG13 | *Botryotinia fuckeliana* | A0A384JFM8 | 31 | Protein glycosylation, lipid metabolic process |
| **6A** | Ribulose bisphosphate carboxylase large chain | *Psychotria* sp. PSN 1 | D6C638 | 118 | Photosynthesis/ photorespiration |
| | Putative oxidoreductase, NAD(P)-binding domain | *Frankia alni* | Q0RJX8 | 33 | Oxidoreductase |
| | Phosphomethylpyrimidine synthase | *Halomonas songnenensis* | A0A2T0V5C0 | 31 | Metabolic process |

**Table 2.** *Cont.*

| Band no. | Protein Name | Plant Species | Accession Number | Protein Score | Biological Function |
|---|---|---|---|---|---|
| **6B** | Signal recognition particle subunit SRP68 | *Klebsormidium nitens* | A0A1Y1IMJ0 | 34 | Transport |
| | Pseudouridine synthase | *Azospirillum brasilense* | A0A560AZY8 | 32 | Ribosome biogenesis |
| | Histidine kinase | *Pseudomonas putida* | A0A7D6A9J0 | 31 | Kinase activity |
| **6C** | Alpha-mannosidase | *Penicillium expansum* | A0A0A2JQF2 | 32 | Metabolic process |
| | Cytochrome c domain-containing protein | *Nitrospirillum amazonense* | A0A560G155 | 31 | Metal binding |
| | Ubiquitin-like domain-containing protein | *Lupinus angustifolius* | A0A1J7GL54 | 31 | Cell cycle |
| | Peroxidase | *Hibiscus syriacus* | A0A6A3AV07 | 28 | Stress response |
| | PNP_UDP_1 domain-containing protein | *Fusarium poae* | A0A1B8A859 | 27 | Metabolic process |
| **6D** | Ribulose bisphosphate carboxylase large chain | *Trichocladus crinitus* | O98531 | 226 | Photosynthesis/ photorespiration |
| | Ribulose bisphosphate carboxylase large chain | *Ceriops tagal* | O20035 | 201 | Photosynthesis/ photorespiration |
| | Ribulose bisphosphate carboxylase large chain | *Trifolium aureum* | A0A023HQ08 | 196 | Photosynthesis/ photorespiration |
| **6E** | Fructose-bisphosphatase | *Brassica napus* | A0A078FJK5 | 45 | Sucrose biosynthesis |
| | Ferredoxin-NADP reductase | *Xanthomonas campestris* | A0A7W6KYF2 | 34 | Oxidoreductase, metal binding |
| | LacI family transcriptional regulator | *Cohnella phaseoli* | A0A3D9INY3 | 33 | Transcription |
| | GDSL esterase/lipase | *Noccaea caerulescens* | A0A1J3D4B7 | 32 | Lipid metabolic process |
| **6F** | Fructose-bisphosphate aldolase | *Spinacia oleracea* | A0A0K9QFF9 | 86 | Glycolytic process |
| | GH43 family beta-xylosidase | *Novosphingobium* sp. *PhB57* | A0A4R3T5L4 | 32 | Carbohydrate metabolic process |
| | Thioesterase domain-containing protein | *Microbispora* sp. *GKU 823* | A0A1V4EJK0 | 32 | Biosynthetic process |
| | Protein kinase domain-containing protein | *Jatropha curcas* | A0A067L8H1 | 32 | Protein kinase activity, ATP binding |
| **6G** | Phosphoinositide 5-phosphatase | *Penicillium italicum* | A0A0A2KL89 | 39 | Lipid metabolic process |
| | Amidase domain-containing protein | *Fusarium poae* | A0A1B8AW33 | 39 | Oxidoreductase |
| | SNF2 domain-containing protein | *Bradyrhizobium huanghuaihaiense* | A0A562QI78 | 36 | ATP binding, helicase activity |
| | Acyl-CoA reductase-like NAD-dependent aldehyde dehydrogenase | *Halomonas stenophila* | A0A7W5EU67 | 36 | Oxidoreductase |
| **6H** | Superoxide dismutase | *Phaseolus lunatus* | Q3S614 | 95 | Stress response, metal ion binding |
| | Putative phospho-2-dehydro-3-deoxyheptonate aldolase | *Phaeomoniella chlamydospora* | A0A0G2EP81 | 32 | Amino acid biosynthesis |
| | Formate dehydrogenase subunit alpha | *Citrobacter freundii* | A0A2S4Q6X5 | 31 | Formate metabolic process |
| | Two-component system alkaline phosphatase synthesis response regulator PhoP | *Staphylococcus* sp. *AtHG25* | A0A318R5E0 | 31 | Transcription |
| **6I** | Ribulose bisphosphate carboxylase small subunit | *Arachis duranensis* | A0A6P4D9J6 | 89 | Photosynthesis/ photorespiration |
| | Epidermal patterning factor-like protein | *Nicotiana tabacum* | A0A1S3YWQ1 | 38 | Cell differentiation, developmental protein |
| | Predicted protein | *Hordeum vulgare* | F2DSS8 | 32 | RNA catabolic process |

**Table 2.** *Cont.*

| Band no. | Protein Name | Plant Species | Accession Number | Protein Score | Biological Function |
|---|---|---|---|---|---|
| **7A** | Ribulose bisphosphate carboxylase large chain | *Phalaenopsis* sp. *SH-2010* | E0D9N8 | 146 | Photosynthesis/ photorespiration |
| | CopA family copper-resistance protein | *Sphingomonas* sp. *BK481* | A0A7W5SGK6 | 49 | Oxidoreductase |
| | Protein TonB | *Bacteroidales bacterium* | A0A7Y5A3N0 | 34 | Protein transport |
| | Replicative DNA helicase | *Candidatus Xiphinematobacter* | A0A0P0FJI7 | 34 | DNA replication |
| **7B** | Ribulose bisphosphate carboxylase large chain | *Kayea stylosa* | Q8MCX9 | 216 | Photosynthesis/ photorespiration |
| | Ribulose bisphosphate carboxylase large chain | *Trifolium aureum* | A0A023HQ08 | 204 | Photosynthesis/ photorespiration |
| | Ribulose bisphosphate carboxylase large chain | *Crossostylis grandiflora* | A0ZQX2 | 204 | Photosynthesis/ photorespiration |
| **7C** | Ribulose bisphosphate carboxylase large chain | *Mucuna* sp. *SH-2010* | E0D986 | 263 | Photosynthesis/ photorespiration |
| | Ribulose bisphosphate carboxylase large chain | *Kayea stylosa* | Q8MCX9 | 262 | Photosynthesis/ photorespiration |
| | Ribulose bisphosphate carboxylase large chain | *Adenophora liliifolioides* | H6VPA5 | 258 | Photosynthesis/ photorespiration |
| **7D** | Cytochrome bo(3) ubiquinol oxidase subunit 1 | *Pseudomonas putida* | A0A059URU4 | 45 | Transmembrane Transport, respiration |
| | 1-phosphatidylinositol 4-kinase | *Cucurbita maxima* | A0A6J1I6A0 | 43 | Lipid metabolic process, signal transduction |
| | Protoporphyrinogen oxidase | *Dothistroma septosporum* | N1PK36 | 32 | Oxidoreductase |
| | Rho-GAP domain-containing protein | *Botryotinia fuckeliana* | A0A384JQ00 | 30 | Signal transduction |
| **7E** | Uncharacterized protein | *Sorghum bicolor* | A0A1B6QGR0 | 37 | Transmembrane transport |
| | Histidine kinase | *Cellulomonas cellasea* | A0A7W4UJ46 | 37 | Signaling |
| | Putative oxidoreductase | *Clavibacter michiganensis* | B0RF25 | 37 | Oxidoreductase |
| **7F** | Fructose-bisphosphate aldolase | *Spinacia oleracea* | A0A0K9QFF9 | 48 | Carbohydrate metabolic process |
| | Ferredoxin-NADP reductase | *Xanthomonas campestris* | A0A7W6KYF2 | 33 | Metal binding, oxidoreductase activity |
| | tRNA(Ile)-lysidine synthetase | *Setosphaeria turcica* | R0JM12 | 31 | tRNA metabolic process |
| | Cyclase family protein | *Streptomyces* sp. *CAI-21* | A0A7Y6LZ28 | 29 | Amino acid metabolic process |
| **7G** | Peptidylprolyl isomerase | *Micractinium conductrix* | A0A2P6V266 | 33 | Isomerase activity |
| | Quinone oxidoreductase, putative | *Colletotrichum orbiculare* | N4V6N5 | 33 | Oxidoreductase |
| | AAHS family benzoate transporter-like MFS transporter | *Arthrobacter* sp. *SLBN-122* | A0A542G607 | 30 | Transmembrane transport |
| | Protein translocase subunit SecE | *Thermobifida cellulosilytica TB100* | A0A147KLB0 | 29 | Protein transport, translocation |
| **7H** | Precorrin-2 dehydrogenase | *Winogradskyella arenosi* | A0A368ZJY7 | 38 | Porphyrin biosynthesis, oxidoreductase |
| | Putative K(+)-stimulated pyrophosphate-energized sodium pump | *Gemmatimonadales bacterium* | A0A7Y4VZE7 | 37 | Sodium ion transport, metal binding |
| | BHLH domain-containing protein | *Physcomitrium patens* | A0A2K1KTX7 | 36 | Transcription |
| | 2,5-diketo-D-gluconate reductase A | *Microbacterium* sp. *SLBN-154* | A0A542N566 | 33 | Oxidoreductase |

**Table 2.** *Cont.*

| Band no. | Protein Name | Plant Species | Accession Number | Protein Score | Biological Function |
|---|---|---|---|---|---|
| **7I** | Uncharacterized protein | *Algoriphagus boseongensis* | A0A4R6T7V4 | 43 | Transmembrane transport |
| | GntR family transcriptional regulator | *Klebsiella quasipneumoniae* | A0A2N4VV92 | 37 | Transcription |
| | Cytochrome P450 | *Mycobacterium* sp. BK558 | A0A4Q7PXL0 | 37 | Oxidoreductase |
| | TonB-dependent receptor plug domain-containing protein | *Nitrospiraceae bacterium* | A0A7Y4SCD5 | 33 | Transmembrane transport |

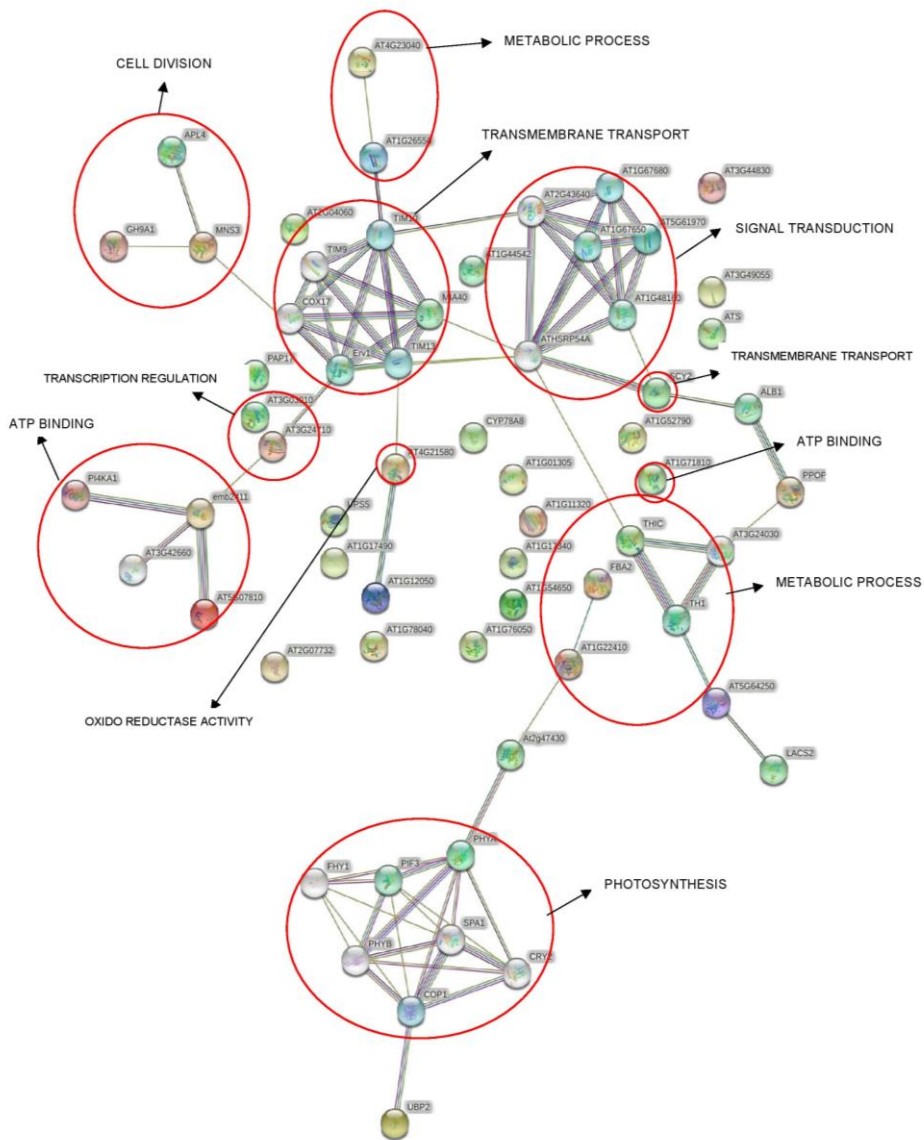

**Figure 10.** Analysis of the protein identified for protein−protein interaction by STRING 9.1 of mung bean (*Vigna radiata*) under Si supply (5 mM) and salinity stress.

### 3.8.3. Proteins Having Oxidoreductase Activity

It was observed that proteins such as ferredoxin-NADP reductase (band 3G, 7F), involved in oxidation/reduction reactions; copA family copper-resistance protein (band 7A), which mediates copper resistance via the sequestration of copper in the periplasm along

with the copper-binding protein CopC; protoporphyrinogen oxidase (band 7D), which is a precursor to heme and chlorophyll; putative oxidoreductase (band 7E); quinone oxidoreductase putative (band 7G); cytochrome P450 (band 7I), which functions as monooxygenase; and 2,5-diketo-D-gluconate reductase A (band 7H) are all downregulated under salinity stress treatments. However, Si supplementation restored the activities of the proteins involved in oxidoreductase activity, such as the putative oxidoreductase NAD(P)-binding domain (band 6A), ferredoxin-NADP reductase (band 6E), amidase domain-containing protein (band 6G), acyl-CoA reductase-like NAD-dependent aldehyde dehydrogenase (band 6G), 2-keto-3-deoxy-L-fuconate dehydrogenase (band 2I), and precorrin-2 dehydrogenase (band 2D).

### 3.8.4. Proteins Involved in Stress Response

Proteins involved in defense responses, such as SOS response UmuD protein (band 3F), which is involved in SOS mutagenesis; diacylglycerol kinase iota (band 3F); and superoxide dismutase (band 3H), were shown to be affected by salinity stress, where the upregulation of stress responsive proteins such as superoxide dismutase (bands 6H and 2H) and peroxidase (band 6C) was witnessed when Si was supplemented.

### 3.8.5. Proteins Responsible for Transmembrane Transport

Transmembrane transport proteins such as DUF4328 domain-containing protein (band 3A); cytochrome b (3); ubiquinol oxidase subunit 1 (band 7D) involved in electron transport; AAHS family benzoate transporter-like MFS transporter (band 7G), which transport a variety of aromatic acids; and cis, cis-muconate, and tonB-dependent receptor plug domain-containing protein (band 7I), which engage in high-affinity binding and energy-dependent uptake with the outer membrane receptor proteins, are downregulated by salinity stress. Meanwhile, Si supplementation upregulated the transmembrane proteins ABC-type branched-subunit amino acid transport system substrate-binding protein (band 2B) involved in amino acid transport, multiple sugar transport system permease protein (band 2C) involved in sugar transport, and acyltransferase (band 2E).

### 3.8.6. Proteins Involved in Signal Transduction

Salinity stress was seen to downregulate proteins such as PAS domain S-box-containing protein (band 3A), which acts as a molecular sensor for sensing redox changes in the electron transport system; 1-phosphatidylinositol 4-kinase (band 7D), which acts as an early signaling system during abiotic stress in plants; Rho-GAP domain-containing protein (band 7D); and histidine kinase (band 7E). However, treatment with Si alone only showed upregulation of the signal transduction protein histidine kinase (band 2H), whose periplasmic domains act as receptor to transduce a signal through its transmembrane domain to the cytoplasmic enzymatic domains.

### 3.8.7. Proteins Involved in Metal Binding

Proteins such as ferredoxin-NADP reductase (band 7F), which acts as the electron acceptor associated with photosystem I, and putative K (+)-stimulated pyrophosphate-energized sodium pump (band 7H), which uses the energy of pyrophosphate hydrolysis as the driving force for Na (+), transport across the membrane. Salinity stress also downregulated the ATPase subunit of the ABC transporter with duplicated ATPase domains (band 3D) and ATP-binding protein (band 3E), and Putative phospho-2-dehydro-3-deoxyheptonate aldolase (band 3H), which were upregulated upon Si supplementation.

### 3.8.8. Proteins Involved in Cell Division

Among the identified proteins, salinity stress was seen to downregulate the protein chromosome partition protein Smc (band 3F). However, silicon treatment alone was seen to upregulate proteins such as TPX2 domain-containing protein (band 2B), which has microtubule binding activity, and shugoshin_C domain-containing protein (band 2F), which

is involved in kinetochore attachment. When Si was supplemented to salinity-stressed plants, upregulation of proteins such as ubiquitin-like domain-containing protein (band 6C) and epidermal patterning factor-like protein (band 6I), involved in cell division and differentiation, were seen to be upregulated.

3.8.9. Proteins Responsible for Transcription and DNA Replication

Salinity stress was seen to downregulate protein kinase domain-containing protein (band 3C) methyltransferase family protein (band 3E), chromosome partition protein Smc (band 3F), SOS response UmuD protein (band 3F), GntR family transcriptional regulator (bands 3H and 7I), and replicative DNA helicase (band 7A), all of which are involved in essential process such as transcription, translation, phosphorylation, DNA replication, etc. Furthermore, Si supplementation enhanced the activities of proteins such as pseudouridine synthase (band 6B) involved in Ribosome biogenesis, LacI family transcriptional regulator (band 6E), and two-component system alkaline phosphatase synthesis response regulator PhoP (band 6H) involved in transcription.

*3.9. Expression of Si-Transporter and Salt-Responsive Genes*

The expression of *Lsi2* and *Lsi3* genes was increased in the salinity- and Si-supplemented groups (Figure 11A–C). The Si-alone treatment showed the highest expression levels of Si transporters, indicating the efficient uptake and transport of Si in mung bean plants under salinity stress. The *Lsi1* gene showed a somewhat increased expression in the salinity plus Si treatment groups. Overall, the expression of Si-transporter genes indicated proficient efflux and influx of Si in plants. Furthermore, salt responsive genes *SOS1* and *SOS3* showed reduced expression levels in the salinity plus Si treatments, whereas the expression of *SOS2* was increased in the salinity plus Si treatment groups (Figure 11C–E).

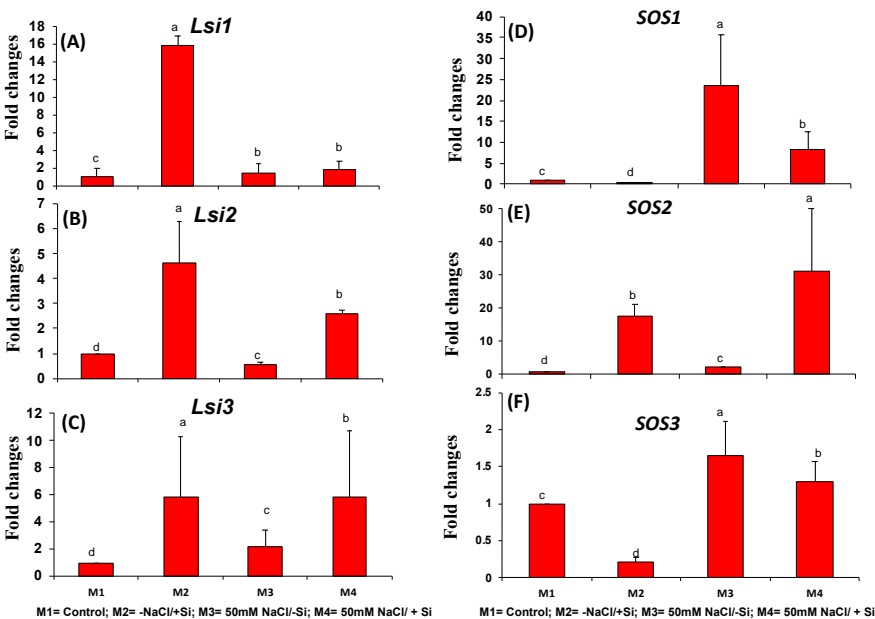

**Figure 11.** The relative expression level of the *Lsi* and *SOS*-related genes: (**A**) *Lsi1*, (**B**) *Lsi2*, (**C**) *Lsi3*, (**D**) *SOS1*, (**E**) *SOS2*, and (**F**) *SOS3* of mung bean (*Vigna radiata*) under Si supply (5 mM) and salinity stress: (a) control (M1), (b) –NaCl + Si (M2), (c) 50 mM NaCl/-Si (M3), (d) 50 mM NaCl/+Si (M4) for a duration of 10 days. Vertical bars indicate mean $\pm$ SE of the mean for $n = 4$. Means denoted by a different letter are significantly different at $p \leq 0.05$ according to Tukey's studentized range test.

## 4. Discussion

Numerous external and internal cues, frequently functioning concurrently, govern the stomata aperture. This comprises of lower soil water potential, water stress-induced

abscisic acid (ABA) generation, and hydrogen peroxide ($H_2O_2$) accumulation in the leaves of plants that are grown in saline soils [38]. Consequently, growth reduction as a result of impairment in photosynthesis may be linked to stomatal closure and the limited $CO_2$ uptake by plants under salinity stress [1]. In our study, we found a similar occurrence of stomatal closure in mung bean exposed to salinity stress, which was also observed in wheat and sweet pepper (Figure 1) [39,40]. However, an efficient regulation of stomatal opening upon Si application hints towards proficient stomatal conductance and $CO_2$ fixation, resulting in superlative photosynthesis leading to proficient plant growth. As far as we are aware, no investigations have been reported on the stomatal opening and closing of plants under salinity stress and Si application; therefore a clear mechanism of Si mediated regulation of stomatal aperture is missing. With the information that ion concentrations of $K^+$, $Cl^-$, and $Ca^{2+}$ mediate the guard cell turgor pressure [41], we may hypothesize that Si application reduced the activity of outward rectifying $K^+$ channels relative to inwardly rectifying $K^+$ channels, and/or reduced $Cl^-$ release from guard cells and lowered the $Ca^{2+}$ concentration inside them, resulting in stomatal opening.

Plants rapidly build up osmotic regulatory substances (such as soluble sugars and soluble protein) in response to abiotic stress to increase the cell-fluid concentration. The primary function of these substances involves the maintenance of cell turgor, balancing protoplasm infiltration and the outside environment, and permitting cells to execute routine physiological processes [42]. In our study, we found that salinity stress inflicted a decline in the levels of soluble protein in a low salinity concentration (10 mM NaCl), which was improved upon Si supplementation. However, a significant impact of Si on increasing the soluble sugar content under salinity was not seen (Figure 2A,B). In wheat plants subjected to salinity stress, Si application improved the soluble protein content significantly, which is assumed to be due to enhance protein kinase synthesis and better cell signaling, which resulted in improved soluble protein levels [43]. Si augmentation increased the soluble protein levels in barley plant leaves and roots [44]. This increase in soluble protein accumulation indicates that the endogenous defense system of plants was strengthened in response to salt stress. Si-mediated regulation of osmolytes has also been reported in alfalfa and basil [45,46], where it has been documented that the movement of osmolytes may progressively provide energy for root development and contribute to the correction of the root's osmotic potential. Thus, Si-mediated accumulation of osmotic regulatory substances in mung bean may be correlated to its enhanced salinity tolerance. Furthermore, secondary metabolites, such as phenolic compounds, are generated in response to adverse environmental circumstances, and are essential for plant growth and reproduction [47]. The buildup of phenolic compounds has been associated with increased ROS scavenging via several mechanisms, including the inhibition of the enzymes involved in ROS generation and quenching [48]. In our study, we observed an increase in the content of phenolic compounds under Si supplementation to intermediate concentrations (20 mM and 50 mM NaCl) of salinity-stressed mung bean plants (Figure 2C). However, the addition of Si to low concentrations of salinity-stressed (10 mM NaCl) plants did not have any considerable impact on the total phenolic content. Comparable observations were also made in alfalfa [45] and tomato [47], where Si supplementation under salinity stress enhanced the plants' phenolic content. The Si-mediated adjustments to the plant secondary metabolism under oxidative stress could increase the phenolic content.

Nitrogen (N) is an indispensable nutrient for plants as it is a building block for many different biomolecules (including proteins, nucleic acids, amino acids, pigments, and hormones) [49]. NR is a substrate-inducing enzyme that is largely active at the transcriptional level and is induced, among other things, by NO3-N, carbohydrates, and light. [50]. However, salinity stress has been shown to deleteriously disturb these progressions, particularly the inorganic uptake of N and the enzymes required in its assimilation into organic compounds, which is not surprising considering the wide range of roles that N plays in plants and the years of study that have gone into understanding the effect of salinity on N metabolism [51]. In cucumber seedlings, growth inhibition instigated by salinity

stress has been linked to changes in nitrogen absorption and enzyme activity involved in nitrogen assimilation [52]. Additionally, salinity stress also inflicts metabolic disorders in compounds of a nitrogenous nature in *Arabidopsis thaliana* [53]. Our study displayed a disruption of NR activity in the roots of mung bean exposed to salinity stress, when compared with the NR activity in leaves. The NR activity in roots was, however, counteracted by Si application to salinity-stressed (10 mM NaCl and 50 mM NaCl) plants, resulting in an improved NR activity in the roots, hinting towards competent N metabolism and nitrogen fixation by mung bean plants (Figure 3A,B). This is in agreement with the findings reported in licorice [54], sunflower [55], and cucumber [56].

Most of the Si research revolves around drawing a relation between Si accumulation and the defensive role it displays in providing abiotic stress tolerance to plants. However, variations in Si accumulation levels among species have been reported and the status of mung bean with regards to efficient Si translocation from roots to shoots remain unexplored. In our study, we aimed to map out the distribution of Si by tracking its accumulation in different parts of the plants. We observed that Si accumulation was affected under salinity stress in both the roots and leaves of mung bean (Figure 4A,B). Furthermore, the accumulation of Si in leaves was slightly greater than the Si accumulation in the roots. These findings suggest a dissimilarity in Si accumulation between the leaves and roots, which is in agreement with the findings in *Glycyrrhiza uralensis* where the accumulation of Si was greater in the shoots than the roots [54]. The deposition of Si on root cell walls, which can inhibit salt transfer to the shoots, may explain why mung bean plants exposed to high salinity experience increased silicon buildup. Additionally, biomass formation aided by Si supplementation may also be a reason for the slightly higher Si accumulation in the shoots [19,20]. The favorable effects of Si were highly associated with the level of Si accumulation in mung bean plants, which could serve as an adaptation strategy for mung bean to decrease salt stress by absorbing and transporting more Si, hence increasing plant growth under salt stress.

A higher concentration of NaCl promotes malfunctioning of the cell membranes, which results in the excessive permeability of ions and electrolytes, which tends to exacerbate oxidative burst in the cells [57]. This was evident by the increase in content of ROS, such as $H_2O_2$ and $O_2^-$, in our study. The $H_2O_2$ content was found to have increased in a low concentration (10 mM NaCl) for the salinity-stressed plants; however Si supplementation could only efficiently scavenge $H_2O_2$ in intermediate concentration (20 mM NaCl) of salinity-stressed mung bean plants. The $O_2^-$ content was also increased upon salinity stress induction. But a competent scavenging of $O_2^-$ was seen under Si supplementation to two concentrations (10 mM and 20 mM NaCl) of salinity-stressed mung bean plants (Figure 5A,B). Our results are in agreement with studies conducted in cucumber [57], rose [20], and wheat [58], where Si conveyed protection against oxidative damage by bolstering the structural integrity of the cell membranes, especially under salt stress.

Several studies have found that stress causes increased reactive oxygen species (ROS) generation, and that supplementing the plant with silicon increases the quantitative changes in the activity of antioxidant enzymes to scavenge these ROS. However, the quantitative shifts on their own are not enough to confirm or depict the intricate changes happening at the cellular level. Protein profiles may change as a result of altered enzyme activity, which may be caused by the downregulation or de novo production of stress-specific antioxidant enzyme proteins [59]. In this regard, not many attempts have been made to demonstrate the changes in isozyme expression profiles, as well as the quantitative changes of the antioxidant enzymes in plants under salinity stress and Si supplementation. However, in our study, a tight regulation of Si in ROS metabolism is demonstrated by Si's constitutive participation in the expression of isozymes of antioxidant enzymes such as SOD, CAT, and APX, which was examined by native-PAGE assay (Figure 6A–D). We observed elevated levels of antioxidant enzyme activity for SOD, CAT, and APX under a low salinity concentration, which was in accordance with the results obtained by Yousif et al. [60] in sorghum, Singh et al. [61] in wheat, and Abdelaal et al. [62] in sweet pepper.

Nonetheless, Shekari et al. [63] reported that Si treatment for herbal *Anethum graveolens* plants under salinity stress significantly increased the activities of CAT, APX, and SOD. Similarly, in rapeseed, Si's participation in increasing the antioxidant enzyme activities was reported by Alam et al. [64]. It is likely that the comprehensive coordination of antioxidant enzymes is indispensable for the redox homeostasis mechanism in mung bean when it is exposed to oxidative stress.

Lignification is a recurrent response of many plant species to several environmental circumstances and mechanical injury, as it reinforces the cell wall for long-distance water transport and gives conducting tissues a structural stiffness and tenacity [65]. The enzymes most directly engaged in lignin production are peroxidases [35]. Previously, it has been demonstrated that salt stress affects secondary cell wall production, as indicated by altered lignification, and that salinity stress is connected with changed anatomical advantageous modifications, such as an increase in lignin deposition in vascular tissues of salt-treated tomato plants and xylem root components in maize [66,67]. However, there are no reports on the role of Si in the lignification of salinity-stressed plants with regards to the expression of peroxidase enzymes. Therefore, in our study, we observed that there was a decreased expression of BPOX, POD, and SPOX in salinity-stressed mung bean, but upon Si supplementation, the expression of these peroxidase enzymes was significantly improved (Figure 7A–C). An increase in the observed peroxidase activity observed may reflect changes in cell wall mechanical properties related to salinity-stress adaption. However, it would be very interesting to find out if there is cross talk between lignin synthesis and Si deposition in salinity-stressed plants, both of which function by providing mechanical protection to the plant cell.

Salinity stress has either direct impacts on photosynthesis, such as stomatal and mesophyll diffusion restrictions and changes in photosynthetic metabolism, or oxidative damage caused by the superimposition of several stressors [68]. Simultaneous stomatal growth redemption and photosynthesis-related protein stimulation indicated Si participation in important carbon fixation processes. Nwugo and Huerta [69] previously demonstrated the augmentation of photosynthesis-related proteins in rice plants exposed to cadmium stress. The primary enzyme required for $CO_2$ fixation during photosynthesis is ribulose-1, 5-bisphosphate carboxylase/oxygenase (RuBisCO). The RuBisCO small subunit (band 6I), in particular, is required for carboxylation catalytic efficiency and $CO_2/O_2$ selectivity. Salinity stress reduced the abundance of RuBisCO large subunit (bands 3C and 3D), whereas Si supplementation increased the abundance of the RuBisCO large subunit (band 6D). Our results are similar to those observed in rose and capsicum under salinity stress and Si supply, where an increased abundance of the RuBisCO small and large subunits was reported [19,20]. Enhanced RuBisCO protein accumulation in Si treatments allowed for photoprotection as well as an improvement in the light-harvesting mechanism for plant physiological growth.

In reaction to abiotic stresses, amino acid, carbohydrate, and amine metabolic pathways undergo various modifications. The activation of early metabolic reactions is essential for cellular adaptation and survival, as it helps correct the chemical and energy imbalances caused by stress. Phosphomethylpyrimidine synthase (band 6A), an important enzyme in pyrimidine biosynthesis, was seen to be upregulated after the addition of Si. A similar protein involved in nucleotide metabolism, adenylosuccinate synthase, was also upregulated when Si was added in capsicum under salinity stress [19]. We have observed that salinity stress downregulated the protein fructose-bisphosphatase (band 3G). However, Si addition upregulated the protein fructose-bisphophatase (band 6E), which is involved in sucrose biosynthesis, indicating that the synthesis and transport of solutes from source to sink have been improved under salinity stress. Under salinity stress, the protein 1-phosphatidylinositol 4-kinase (band 7D), an essential enzyme in the salicylic acid mediated signal transduction pathway is downregulated, but the addition of Si was found to have upregulated the protein lipase (band 6E). The large majority of lipid-associated plant defense responses are facilitated by lipase activation that cleave or modify lipid

substrates in several subcellular compartments (Lee et al., 2019). Thus, Si supplementation promotes effective regulation of the metabolic processes in plants, allowing them to cope with stress [70]. Thus, Si supplementation mediates the efficient regulation of metabolic processes in the plants, such that the plants can cope up with stress.

During oxidative stress, ROS production is often significantly elevated. The excessive ROS production under conditions of salt stress primarily depletes vital metabolic pathways and protein synthesis [18]. The superoxide dismutase (SODs) is the first line of defense against ROS within a cell [71]. We found the upregulation of SOD (band 2H) after Si supplementation alone as well as when Si was supplemented to salinity-stressed plants (band 6H). Similarly, Si supplementation to salinity-stressed plants also upregulated peroxidases (band 6C), whose functions in ROS scavenging, lignin biosynthesis, and the consequent defense response activity in plants have been well documented. Plant peroxygenase is generally engaged in the $H_2O_2$-dependent hydroxyl catalysis of aromatics, sulfoxidation of xenobiotics, and oxidation of unsaturated fatty acids [72]. Our results are in agreement with studies conducted in rice under Cd stress [73], alfalfa under salt stress [15], and rose under salinity stress and Si supply [20], where stress-induced activation of peroxidases and its protective role have been documented. Thus, the presence of a competent balance between ROS formation and its scavenging helps mung bean in ameliorating the negative effects of salinity stress.

Under abiotic stress conditions, plant tissues endure substantial oxygen variations, resulting in a highly hypoxic environment [74]. Furthermore, ROS exposure produces an oxidative environment that alters the redox equilibrium of the cell. As numerous intracellular signaling pathways governing cell division and stress response systems are sensitive to redox conditions, intracellular alterations in redox status also have a significant impact on cell functioning [75]. Several physiological activities, including redox activity, are mediated by electron-transporting oxidoreductases in biological membranes. In our study, we found that salinity stress downregulated the protein Ferredoxin-NADP reductase (band 3G) whereas Si application lead to an upregulation of Ferredoxin-NADP reductase (bands 6A and 6E). During the linear electron transport process of photosynthesis, ferredoxin, NADP (H) oxidoreductase (FNR) transports electrons from ferredoxin (Fd) to $NADP^+$. For reductive assimilation and light/dark activation/inactivation of enzymes, both NADPH and reduced Fd ($Fd^{red}$) are essential [76]. As a result, FNR acts as a hub that connects electron transport in photosynthesis to redox metabolism in chloroplasts, and Si supplementation positively affects the redox balance in the plants. Reactive aldehydes formed as a by-product of lipid peroxidation are detoxified by reducing their carbonyl group to alcohol or oxidizing it to the corresponding carboxylic acid [77]. This oxidative reaction is known to be catalyzed by $NAD(P)^+$-dependent aldehyde dehydrogenases (ALDHs), whose accumulation was improved under Si supplementation (band 6G); this was similar to the findings on alfalfa under salt stress, where alcohol dehydrogenase activity involved during oxygen deprivation was increased [15]. Our results are also consistent with the findings on redox homeostasis of rose and tomato under salinity stress and Si supplementation [18,20].

The degree to which plants profit from Si depends on its deposition in the tissues, which normally ranges from 0.1% to 10% (by dry weight) and exhibits significant cultivar, species, and wider evolutionary variations [10]. This is made conceivable by the various Si-transporter genes that mediate the coordinated uptake and distribution of Si along the plant parts. The plasma-membrane transporters encoded by these genes are hypothesized to coordinate the symplastic migration of Si to circumvent the apoplastic (casparian band) barriers [78]. The identification of *OsLsi1* and *OsLsi2* in rice [14], *ZmLsi1* and *ZmLsi2* in maize [79], *CsLsi1* and *CsLsi2* in cucumber [80], and *HvLsi1* and *HvLsi2* in barley [81], has enlightened the research community about the uptake and distribution of Si in plants and its relation with the suppressed or enhanced expression of *Lsi* (influx and efflux) genes in providing ameliorative benefits to plants under various biotic and abiotic stresses. To the best of our knowledge, the role of Si transporters in mung bean under salinity stress has not been explored before. Consequently, we observed that even under salt stress, *Lsi1*, *Lsi2*, and

*Lsi3* were considerably expressed in Si-treated plants; however under salt stress without Si treatment, the expression levels were lower (Figure 11A–C). Our results are in line with Muneer and Jeong [18], who illustrated an increased expression of *LeLsi1*, *LeLsi2*, and *LeLsi3* genes in tomato under salinity stress and Si supplementation. This synergistic activation of silicon transporter genes under salt stress implies a role for Si in salt-stress mitigation. The expression of Si transporters is regulated differently in different plant species. The mechanisms that modulate Si transporter gene expression, however, remain unclear.

The SOS pathway is seen as crucial for managing both $Na^+$ efflux out of the root cortex, as well as long-distance transport into the plant tissue via the xylem [82]. SOS machinery, consisting of *SOS1*, *SOS2*, and *SOS3* proteins, were extensively studied in an effort to comprehend the process of ion homeostasis and salinity tolerance in a vast array of plant species [83]. In *Arabidopsis*, sos1 mutants subjected to moderate salinity displayed decreased Na accumulation in the leaves, indicating *SOS1* involvement in $Na^+$ xylem loading [84]. In salinity-stressed maize plants supplemented with Si, an increased expression of *SOS1* and *SOS2* genes was observed along with an increase in the $Na^+$ in the xylem and leaf tissues [85]. Transgenic rice overexpressing *SOS1* have a significantly higher salt tolerance than wild-type rice when supplemented with Si. This improvement in tolerance is coupled with enhanced $Na^+$ efflux in transgenic roots and greater $K^+$ absorption, resulting in less $Na^+$ buildup in cells and a higher $K^+/Na^+$ ratio [86]. This is in agreement with our results where we observed an increased expression of *SOS1* and *SOS2* in mung bean plants under 50 mM of salinity stress supplemented with Si, indicating competent $Na^+$ efflux from the cell (Figure 11D–F). *SOS3* is known to be a $Ca^{2+}$ regulated SOS pathway upstream regulatory protein that plays critical roles in pathways related to salt-stress response [87]. Kim et al. [88] demonstrated that NaCl treatment induces *At*SOS3 expression significantly. Moreover, the overexpression of *Le*SOS3-1 improves salt-stress tolerance in tobacco through the regulation of stress-related physiological changes. Similarly, we observed an increased expression of *SOS3* under Si supplementation in mung bean under salinity stress, indicating the efficient phosphorylation of *SOS1*, thus leading to competent efflux of $Na^+$ ions from the cells. This is in accordance with another study on sugarcane, where, in response to salt stress, most *VviSOS3* genes were regulated similarly in all three organs. These findings imply that *VviSOS3* genes may play a role in ionic and osmotic homeostasis establishment and maintenance [89]. Thus, it can be concluded that the Si-mediated expression of SOS genes leads to the sequential efflux, compartmentalization, and blockage of $Na^+$ ions in mung bean plants, thus retaining optimum levels of beneficial ions, which allow the cells to remain viable and hence assist towards tissue growth even under salt stress. The precise methods through which Si assists SOS gene transcription remain unknown; one can only hypothesize that Si indirectly impacts transcription factors, but additional study is required.

## 5. Conclusions

In conclusion, Si-mediated regulation of stomatal aperture, osmoregulatory substances, N metabolism, and ROS homeostasis, provided the physiochemical basis of salinity stress tolerance in mung bean. Although soluble protein and phenolic content were seen to be enhanced under Si supplementation, a convincing improvement in soluble sugar content upon Si supplementation was missing. Similarly, the increase in root NR activity after Si supplementation also shed light on the active N uptake and its metabolism. Antioxidant defense against oxidative-stress-induced damage was maintained by antioxidant enzymes and their isozyme activity. We observed that isozymes and antioxidant molecules appeared to play a significant role in providing competent defense to plants against salinity. Furthermore, the dynamic role of Si in the regulation of proteins engaged in diverse cellular functions and metabolic pathways may benefit from a better understanding of the potential mechanism(s) evolved in plants to ameliorate the adverse effects of salinity stress. Gene expression studies of Si transporter and salt responsive genes revealed an optimum expression of these genes in response to salinity stress and Si supplementation. An optimal expression of Si transporters in plants with inefficient Si absorption systems may result in

an increase in Si accumulation, hence boosting the plants' resilience to numerous stressors. However, investigating probable interaction partners and expression patterns of Si transporter genes will lend evidence to their likely roles in plant stress responses. Our research brings up the possibility of using Si transporters for breeding objectives. However, the roles of this transporter in Si fluxes and plant physiology in general must be clarified if we are to successfully use Si as a preventive measure against environmental stress.

**Author Contributions:** S.M. conceptualized and supervised the work; M.A.M. performed the experiments and wrote the manuscript. S.M. edited and finalized the manuscript. All authors have read and agreed to the published version of the manuscript.

**Funding:** The research was funded by VIT Seed Grant (Project no. SG20210182).

**Institutional Review Board Statement:** Not applicable.

**Data Availability Statement:** The data presented in this study are available on request from the corresponding author. The data are not publicly available due to privacy concerns.

**Acknowledgments:** The authors would like to thank Central Instrumentation Facility, Gyeongsang National University, South Korea for mass spectrometer analysis. The authors would also like to acknowledge Scanning Electron Microscopy Facility, School of Biosciences and Technology, Vellore Institute of Technology for helping carry out the SEM micrographs.

**Conflicts of Interest:** Authors declare no conflict of interest.

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
