# Peer review of "Physiological and Molecular Analysis Revealed the Role of Silicon in Modulating Salinity Stress in Mung Bean"

_agriculture, doi:10.3390/agriculture13081493_

Round 1

Reviewer 1 Report

General comments:

The manuscript presented by Almurad and Muneer investigated the physiological and molecular analysis revealed the role of Silicon in modulating salinity stress in mung bean. Some of my general and specific comments are below:

First of all, I can't use line numbers as they aren't there. The second reference used in this manuscript was about a combination between gas and fertilizer as a source on nutrition. Moreover, the study conducted in 2010. So, I don't think it's a good one to use it proven that farmers use as 10 time as the requested amount of fertilizer in general. Reference number four was about using plant regulators in traditional Chinese medicine not vegetables and fruits affected by excessive fertilizer applications.

Specific comments:

  • Line 10: Add the scientific name of the mug bean.
  • Line 31-34: Climate change has nothing to do with water quality!!! Reword this sentence and focus on the issue of water quality in some regions and remove the impact of climate change since you are not focusing on heat and temperature facts.
  • Line 35: Reorder the impact of salinity on plants; for instant "as it damages cell membranes, reduce the osmotic potential ….." to reduce the osmotic potential which cause ……. At the end led to damages cell membranes". So, used whatever you like but keep it in order.  

Writing style is good

Author Response

Response sheet

#Reviewer 1

First of all, I can't use line numbers as they aren't there. The second reference used in this manuscript was about a combination between gas and fertilizer as a source on nutrition. Moreover, the study conducted in 2010. So, I don't think it's a good one to use it proven that farmers use as 10 time as the requested amount of fertilizer in general. Reference number four was about using plant regulators in traditional Chinese medicine not vegetables and fruits affected by excessive fertilizer applications.

 Response >

Authors thank the reviewer for their review.

However, Authors would like to state that the comment mentioned about to change ‘second reference’ is not clear. Because our second reference is not from ‘2010’, but the reviewer has mentioned to change it as it is from ‘2010’. So authors request the reviewer to please clarify which reference to be changed.

The authors have changed the mistake regarding reference four as suggested by the reviewer.

Line numbers in manuscript will be automatically given by the journal and is already there. Some Microsoft office does not support the line numbers given by journal software. Thus you might not be able to see line numbers.

Specific comments:

Line 10: Add the scientific name of the mug bean.

Response >

The scientific name of mung bean has been added in the specific line as mentioned.

Line 31-34: Climate change has nothing to do with water quality!!! Reword this sentence and focus on the issue of water quality in some regions and remove the impact of climate change since you are not focusing on heat and temperature facts.

Response >

Authors agree with the reviewer and have made the necessary corrections.

Line 35: Reorder the impact of salinity on plants; for instant "as it damages cell membranes, reduce the osmotic potential ….." to reduce the osmotic potential which cause ……. At the end led to damages cell membranes". So, used whatever you like but keep it in order. 

Response >

Authors agree with the reviewer and have made the necessary corrections regarding the order of impact of salinity in plant growth and development.

Reviewer 2 Report

The manuscript titled “Physiological and Molecular Analysis Revealed the role of silicon in Modulating Salinity Stress in Mung Bean” The study focused on the effects of different salinity concentrations on antioxidant capacity, proteome level alterations, and influence on Si-transporter and salt responsive genes. This study brings the mechanism behind protection against excessive ROS, connect the various dots related to changes in protein expression under salinity stress and Si supplementation using LC-MS/MS, and examine the role of mRNA level regulation of Si transporter genes and salt responsive genes in mung bean under salinity stress and Si supplementation. However, some revisions are necessary so that this work may be considered for publication after the required revision.

There are much needed importance of clarity in abstract and introduction with fluency of research design and most importantly, discussion portion is also needing major revision to link it with your findings and ground reality based results of previous researches.

Language is just a medium that can convey your message spanning life and importance moments of your life to find special findings, so I emphasize the authors to improve English language and grammar of article and try to make explanation simple as much as you can to attract new and non-familiar or non-native readers.

Literature citation is one of great importance, in this article several papers are of older version, so try to replace recently published articles related to this research such as “DOI: 10.3390/antiox11020359, DOI: 10.1093/plphys/kiad317”. Evaluate the results and link them all in complimentary manner so that a clear picture draws your efforts for solving the problem in this research domain.

Minor suggestions

Abstract

The abstract is short, try to explain a little bit about the results in the abstract like how much percentage of functional classification of proteins, what genes were observed and what their regulation under salt and Si stress were 

Introduction

Too much information can reduce the introduction and be specific

M&M Section:
Pg 5, L27: Were the harvested leaf samples snapped-frozen in Liquid N2 prior to storage in -80 degrees C?
Pg 6, L16-17: Were the leaf samples squashed using mortar and pestle? Whose protocol was used?
The authors should be consistent in using rpm or rcf as the centrifugation unit. g stands for gravity and needs to be italicized.
Reference Giannopolitis and Ries, 1977 did not explain how SOD and NBT assays were done.
What are the modifications and changes made in APX assay? How was percent inhibition and SOD concentration calculated? What about APX activity? How was it calculated?
First-dimension Proteomics should separate total proteome based on pH by isoelectric focusing (IEF) not sizes using SDS-PAGE. I am afraid what the authors were doing is just SDS-PAGE protein analysis and protein Identification via Mass Spectrometry and not Proteomics. Specify the reasons why SDS-PAGE was used instead of 2D 
how were the proteins identified? Using which database? How was the peptides quantified? What software was used to analyse peptide data? How did you define the significance changes across treatments? How was protein-protein interaction analysis done?
RT-PCR was done using SYBR green chemistry. This method is less sensitive compared to TaqMan probes. Why the former was chosen for the study was not justified.
Result Section
I could not fathom how the percentage calculation for increase and decrease was made throughout the manuscript. Mention in the statically analysis section
Improve the quality of gel pictures

You can make string analysis as another figure so that it is visible (Figure 10) and RT-PCR can be Figure 11

Bring your table T1 as the main table 

Discussion

Discussion portion is also needing major revision to link it with your findings and ground reality based results of previous researches.

Language is just a medium that can convey your message spanning life and importance moments of your life to find special findings, so I emphasize the authors to improve English language and grammar of article and try to make explanation simple as much as you can to attract new and non-familiar or non-native readers

Author Response

#Reviewer 2

There are much needed importance of clarity in abstract and introduction with fluency of research design and most importantly, discussion portion is also needing major revision to link it with your findings and ground reality based results of previous researches.

Response>

Authors have considered the reviewers suggestions and incorporated all the changes. Please check the highlighted portions in the revised text

Language is just a medium that can convey your message spanning life and importance moments of your life to find special findings, so I emphasize the authors to improve English language and grammar of article and try to make explanation simple as much as you can to attract new and non-familiar or non-native readers.

Response>

Authors have considered the reviewers suggestions and have improved the English and grammatical errors using a licensed software of the institute (quillbot)

Literature citation is one of great importance, in this article several papers are of older version, so try to replace recently published articles related to this research such as “DOI: 10.3390/antiox11020359, DOI: 10.1093/plphys/kiad317”. Evaluate the results and link them all in complimentary manner so that a clear picture draws your efforts for solving the problem in this research domain.

Response>

Authors have added all the recent references suggested by the reviewer wherever necessary. Please refer your comment given at the end

Abstract

The abstract is short, try to explain a little bit about the results in the abstract like how much percentage of functional classification of proteins, what genes were observed and what their regulation under salt and Si stress were 

Response >

The Authors have taken up the reviewer’s suggestion and have thus, added more elaborate information about the findings of the research in the abstract. Please see the highlighted areas of abstract section

Introduction

Too much information can reduce the introduction and be specific

Response >

Authors agree with the reviewer and have cut short the introduction section.

M&M Section:

Pg 5, L27: Were the harvested leaf samples snapped-frozen in Liquid N2 prior to storage in -80 degrees C?

Response >

Yes, the leaves were snapped-frozen in Liquid N2 prior to storage in -80ºC

Pg 6, L16-17: Were the leaf samples squashed using mortar and pestle? Whose protocol was used?
The authors should be consistent in using rpm or rcf as the centrifugation unit. g stands for gravity and needs to be italicized.

Response >

Yes, the leaves were squashed/homogenized using mortar and pestle.

The authors have made the necessary corrections and have uniformly used ‘rpm’ throughout the manuscript to refer for centrifugation speed.

Reference Giannopolitis and Ries, 1977 did not explain how SOD and NBT assays were done.

Response >

The authors would like to convey that, it was Giannopolitis and Ries, (1977), who in their article ‘Superoxide Dismutases: I. occurrence in Higher Plants’, published in journal ‘Plant Physiology’, explained and performed the SOD activity in shoots, roots and seeds of corn, oats and peas which served as the basis of determination of SOD activity in plants by NBT inhibition method, henceforth. Kindly check.

What are the modifications and changes made in APX assay? How was percent inhibition and SOD concentration calculated? What about APX activity? How was it calculated?

Response >

The protocol that was followed for APX activity including the modification is given below. We are not including the detailed protocol in the text so as to avoid plagiarism issues.

For determining APX activity, 0.3 g of tissues were homogenized in 3 mL of extraction buffer (KH2PO4, K2HPO4, 1% PVP, 1% TritonX-100, and EDTA) and 1 mL of 5 mM ascorbate. The samples were centrifuged for 20 min at 10,000 rpm at 4 °C and the resulting supernatant was mixed with 2 mL of reaction buffer containing KH2PO4, and K2HPO4 (pH7.3). The absorbance was determined at 290 nm with 30 s intervals for 3 min (E = 2.8 mM−1cm−1).

The Sod activity was calculated using the following formulae:

SOD activity= [{(Abs. reading) x (dilution factor) x 60} / {50 x (Protein content) x (Incubation time)}] X 100

The APX activity was calculated using the following formulae.

APX activity= (vol. of assay x Abs. reading) / (extinction coefficient x vol. of enzyme x protein content)

First-dimension Proteomics should separate total proteome based on pH by isoelectric focusing (IEF) not sizes using SDS-PAGE. I am afraid what the authors were doing is just SDS-PAGE protein analysis and protein Identification via Mass Spectrometry and not Proteomics. Specify the reasons why SDS-PAGE was used instead of 2D 

Response >

For many years 2D-PAGE was the method of choice for protein-level fractionation prior to MS analysis. The approach has the capacity for high resolution based on two orthogonal physical properties of a protein and the ability to focus identification efforts (albeit only for more abundant proteins). However, the method is poorly suited for more global protein identification. Although many techniques have been developed to improve the quality and number of spots in a 2D gel, these enhancements are still not good enough to study the entire cellular proteome. Thus, 2-DE has lately become a debatable method due to its known limitations and in part to the development of alternative MS-based approaches. Some of the reasons behind this trend include issues related to reproducibility, poor representation of low abundant proteins, highly acidic/basic proteins, or proteins with extreme size or hydrophobicity, and difficulties in automation of the gel-based techniques.

Fractionating samples at the protein level is one of the most common ways to circumvent challenges due to sample complexity and improve proteome coverage. In this regard, one-dimensional sodium dodecyl sulfate-polyacrylamide gel electrophoresis followed by liquid chromatography-tandem mass spectrometry (GeLC-MS/MS) is a robust and reproducible method for qualitative and quantitative proteomic analysis, and thus we have used this approach for our research. The 1D SDS-PAGE method is a logical choice for a wide range of protein samples due to high resolving power, protein capacities in excess of sample loads for modern nLC-MS/MS instruments, compatibility with many detergents and chaotropes used in sample extraction, cost, and availability. The combination of protein-level separation by 1D-SDS-PAGE followed by RP LC-MS/MS analysis of digests from all bands, referred to as GeLC-MS/MS, offers a powerful analytical approach that balances real-world constraints with obtaining optimal proteome coverage.

how were the proteins identified? Using which database? How was the peptides quantified? What software was used to analyse peptide data? How did you define the significance changes across treatments? How was protein-protein interaction analysis done?

Response >

The MS and MS/MS spectra data were analyzed with a mass tolerance of 50 ppm using NCBI and Protein Pilot V.3.0 database software (with the MASCOT V.2.3.02 database search engine). Oxidation of methionines and carbamidomethylation of cysteines were permitted for database searches of MS/MS spectra. Individual peptide ion scores were searched using a statistically significant threshold value of p = 0.05. According to gene ontology (http://www.geneontology.org), the identified proteins were categorized based on the biological processes in which they contribute. The identified proteins were checked for any protein-protein interactions using STRING data base

RT-PCR was done using SYBR green chemistry. This method is less sensitive compared to TaqMan probes. Why the former was chosen for the study was not justified.

Response >

The Authors agree with the reviewer that TaqMan probes have higher specificity than SYBR green chemistry. However, SYBR green chemistry was used in our research. SYBR green is appropriate for qualitative and quantitative detection, but careful design and some optimization is needed to obtain a good assay. SYBR green, like all other intercalating dyes, binds any double-stranded DNA, so in order to separate non-specific from specific amplicons, it must be assumed that different PCR products will have different melting temperatures. Under this assumption, a so-called melting (“dissociation”) curve can be generated by carefully monitoring the fluorescence properties of the PCR amplification products during a melting phase. This might aid the user in ensuring that non-specific amplification has not taken place. The main advantage of using SYBR Green for RT-PCR is that there is no requirement for the incorporation of a fluorescent reporter system into the primer design or the synthesis of fluorescently labeled probes or beacons that are specific only for a target sequence. Also, SYBR green is cost effective and easy to use. Moreover, with proper optimization in primer designing such that they are closely matched in Tm, calculate or estimate approximate Tm, using standard hot-start PCR mixes, almost similar results can be obtained with SYBR green chemistry as with TaqMan probes
ResultSection
I could not fathom how the percentage calculation for increase and decrease was made throughout the manuscript. Mention in the statically analysis section
Improve the quality of gel pictures

Response >

As per reviewer’s suggestion, Authors have mentioned how percentage calculation was made throughout the manuscript. During galley proof we will provide original picture that can be used for high quality

You can make string analysis as another figure so that it is visible (Figure 10) and RT-PCR can be Figure 11

Response >

The Authors have followed the reviewer’s suggestion and changed string analysis into a single figure. And have also changed the number for RT-PCR figure.

Bring your table T1 as the main table 

Response >

The table T1 has been incorporated in the main text as Table 1.

Discussion

Discussion portion is also needing major revision to link it with your findings and ground reality based results of previous researches.

Response >

The Authors have followed the reviewer’s suggestion and have revised the discussion section along with replacing old references with new one such as Auoz et al., 2023 ; Alam et al., 2023 ; Singh et al., 2022 ; Ahmad et al., 2022; Akhter et al., 2022 ; Yousif et al., 2021 etc.

Reviewer 3 Report

In general, the research is interesting. However, from the manuscript I have concerned many points.

The authors studies different salinity concentrations on physiological responses of Mung Bean, but in the results, the authors did not explain or describe the effect of different concentration of NaCl on those physiological responses, also protein and gene expression. I feel that the authors miss some important points from their results, and do not use the result from their statistical analysis, but only focus some points for the reader to follow. The letters of statical analysis on many graphs are concerned, and needed to be checked. My comments to improve are below.

 Materials and methods

1. Experimental design, when were seedlings treated with NaCl? Was it treated from germination to 10 days of age? Please clarify

2. Lines 281-282, the authors wrote that ‘For physiological parameters complete randomized design was utilized with three replicates”. However, the figure legends of Figures 2, 3, 4, 5, 6, and 10 showed that “Vertical bars indicate Mean±SE of the means for n = 4”. What is the difference between three replicates and n= 4? Please explain the details in each analysis method for the reader.

 Results

1. Lines 301-313, please explain more details with statistical analysis of the results from Figure 2A, B, and C.

-        Lines 303-306, I see as Si supplementation significantly increased total soluble sugar for 10 mM NaCl, but significantly reduced in 20 mM NaCl, and did not affect for 50 mM NaCl.

-        Lines 306-309, the authors wrote that “Similarly, the total soluble sugar content in salinity treated plants was seen to be increased upon Si supplementation by 53% and 37% in T6 and T8 respectively when compared to T5 & T7 respectively (Figure 2B)”. Please correct the sentence because the results showed no statistical differences as the reader will slightly get the wrong point. From the results I can see that there is no effect of Si on total soluble sugar. 

-        For total phenolic content, Si did not significantly increase it at 10 mM NaCl, but significantly increased for 20 and 50 mM NaCl.

2. Lines 321-330, please explain more details with statistical analysis of the results from Figure 3A, and B.

- Lines 322-324, the authors wrote “Salinity stress was seen to affect the NR activity in leaves as well as roots, but Si supplementation ameliorated the salinity stress and restored the NR levels back to normal (Figures 3A and 3B)”, but Figure 3A showed that only 10 mM NaCl significantly reduced leaf NR activities, but 20 and 50 mM NaCl did not significantly affect NR activities. Moreover, Figure 3B showed that 10, 20 and 50 mM NaCl did not significantly affect root NR activities.

 3. Lines 338-350,

- the unit of Figure 4 should be changed to “µg/ gDW or µg/mgDW”, please change.

- Figure 4A, please check the letter on bars for a, b, d, e (T3= e?, T5 = a?, but no “c” on the graph)

 - Figure 4B, please check the letter on bars for a, b, d, e (T3= e?, T5 = a?, but no “c” on the graph)

- Lines 339-340, the authors wrote “In leaves, the highest Si content was observed in Si alone treatment (T2) (Figure 4A & 4B)”; it is not true because 50 mM NaCl + Si (T8) showed significantly higher than Si alone (T2), and please delete (Figure 4B).

- Lines 340-341, the author wrote “Si content increased in T4, T6 & T8 by 21%, 49% and 81% respectively”, as compared to……………(if the authors check statical analysis, they may be not significantly increased, please check).

- Lines 342-343, please explain more details of the results first.

 4. Lines 351-367

- Lines 353-354, please explain the results of T5, T7

- Lines 358-360, what is about the result of T8, please explain.  

 5. Lines 368-399

- Figure 6A, please check the letter on bars for a, b, c, d, e (T3= de?, T5 = bcd?, T7 = e?)

- Figure 6A, please define the unit of enzyme activity in the methods for U mg protein/minute.

- Lines 370-371,..with the exception of T3” what does the author mean?

- Lines 371-372, the authors wrote “Si supplementation was however seen to enhance the activity of SOD by 185% and 101% in T6 and T8 respectively when compared to T5 and T7 respectively.”, but the results showed T5 and T6 are not significantly different, please consider.

- Lines 378-379, the results are not significantly different, please consider (Figure 6B).

- Lines 382-385, the results are not significantly different, please consider (Figure 6C).

 6. Lines 441-446, can Si supplementation positively influence RuBP in every salinity treatment, please provide data.

 7. Lines 535-551

- Figure 10 is not eight treatments, it is only four, please correct.

- Figure 10, it is good to use T1,T2….., not M1, M2,…

 8. Expression of Si-transporter and salt responsive genes

- Lines 536-543, the authors wrote “The expression of Lsi2 and Lsi3 genes was increased in salinity and Si supplemented groups (Figure 10A, 10B and 10C). However, the results show no significant different between NaCl and NaCl+Si. So, I disagree with the authors as it is the key message for the reader.

Moreover, Si application also did not significantly change SOS1,2, and 3 as compared between NaCl and NaCl+Si.

 Discussion

1. Lines 576-579, please consider, see my comments for results

2. Lines 587-589, please consider, see my comments for results

3. Lines 604-608, please consider, see my comments for results

4. Lines 616-618, please consider, see my comments for results.

Also it is difficult to conclude that Si in leaves is greater than in root as the authors did not statistically analyse, I suggest analyse them.

5. Lines 630-632, please consider, see my comments for results.

6. Lines 763-766, please consider, see my comments for results. (What are VrLis1, VrLis2, VrLis3?, They are not presented in Figure 10).   

7. Lines 784-787, please consider, see my comments for results.

8. Lines 798-800, I disagree, please see my comments for results

 Conclusion

-        Too long, please concise, and consider my comments for results and discussion.  

-         Abstract

-        Please rewrite with more focus of statical analysis results

References

-        Please follow the journal template as [number]

-

Author Response

#Reviewer 3

 Materials and methods

  1. Experimental design, when were seedlings treated with NaCl? Was it treated from germination to 10 days of age? Please clarify

Response >

The Authors have treatment mung bean plants with NaCl after the plants had reached the vegetative stage. This is mentioned in the methodology section

  1. Lines 281-282, the authors wrote that ‘For physiological parameters complete randomized design was utilized with three replicates”. However, the figure legends of Figures 2, 3, 4, 5, 6, and 10 showed that “Vertical bars indicate Mean±SE of the means for n = 4”. What is the difference between three replicates and n= 4? Please explain the details in each analysis method for the reader.

Response >

The authors have made a typo error in Lines 281-282. Instead of writing ‘four replicates’ the authors have written ‘three replicates’. Thus, four replicates were used in the experiments represented in Fig, 2,3,4,5,6 and 10 , hence n=4 was written in the figure legend. Authors have corrected the same in the text.

  1. Lines 301-313, please explain more details with statistical analysis of the results from Figure 2A, B, and C.

Response >

Authors have re-written the results for Figures 2A, B and C as suggested by the reviewer with emphasis on statistical analysis.

Lines 303-306, I see as Si supplementation significantly increased total soluble protein for 10 mM NaCl, but significantly reduced in 20 mM NaCl, and did not affect for 50 mM NaCl.

Response >

The authors agree with the reviewer that Si supplementation significantly increased total soluble protein for 10 mM NaCl, but significantly reduced in 20 mM NaCl, and did not affect for 50 mM NaCl, as it is evident from the statistical analysis. Hence, authors have rewritten the result for total soluble protein content and its change upon Si supplementation.

        Lines 306-309, the authors wrote that “Similarly, the total soluble sugar content in salinity treated plants was seen to be increased upon Si supplementation by 53% and 37% in T6 and T8 respectively when compared to T5 & T7 respectively (Figure 2B)”. Please correct the sentence because the results showed no statistical differences as the reader will slightly get the wrong point. From the results I can see that there is no effect of Si on total soluble sugar. 

Response >

The Authors agree with the reviewer regarding the disparity between statistical significance and written text. Therefore, the authors have re-written the result for total soluble sugar as suggested so that the correct message is conveyed to the readers.

      For total phenolic content, Si did not significantly increase it at 10 mM NaCl, but significantly increased for 20 and 50 mM NaCl.

Response >

The authors have re-written the results and have added that Si supplementation did not significantly increase phenolic content in T4 (10Mm NaCl) as is evident from the statistical analysis.

  1. Lines 321-330, please explain more details with statistical analysis of the results from Figure 3A, and B.

Response >

As suggested by the reviewer the authors have re-written the results for NR activity with emphasis on statistical significance. Please check.

Lines 322-324, the authors wrote “Salinity stress was seen to affect the NR activity in leaves as well as roots, but Si supplementation ameliorated the salinity stress and restored the NR levels back to normal (Figures 3A and 3B)”, but Figure 3A showed that only 10 mM NaCl significantly reduced leaf NR activities, but 20 and 50 mM NaCl did not significantly affect NR activities. Moreover, Figure 3B showed that 10, 20 and 50 mM NaCl did not significantly affect root NR activities.

Response >

The authors agree with the reviewer and thus after checking the statistical significances, the authors have re-written the sentence so that there is correct depiction of the results with respect to statistical analysis.

  1. Lines 338-350,

The unit of Figure 4 should be changed to “µg/ gDW or µg/mgDW”, please change.

Response >

The unit for Si content has been changed as asked by the reviewer

Figure 4A, please check the letter on bars for a, b, d, e (T3= e? T5 = a? but no “c” on the graph)

Response >

The authors have re-analyzed the data for mistakes in letters placed on bars and have rectified the error in figure 4A

Figure 4B, please check the letter on bars for a, b, d, e (T3= e? T5 = a?, but no “c” on the graph)

Response >

The authors have re-analyzed the data for mistakes in letters placed on bars and have rectified the error in figure 4B

Lines 339-340, the authors wrote “In leaves, the highest Si content was observed in Si alone treatment (T2) (Figure 4A & 4B)”; it is not true because 50 mM NaCl + Si (T8) showed significantly higher than Si alone (T2), and please delete (Figure 4B).

Response >

The authors have modified the sentence according to the reviewer’s query. T8 showed higher Si content thus it has been included in the result as well.

Lines 340-341, the author wrote “Si content increased in T4, T6 & T8 by 21%, 49% and 81% respectively”, as compared to……………(if the authors check statistical analysis, they may be not significantly increased, please check).

Response >

After the authors have re-analyzed the results for statistical errors, now the line “Si content increased in T4, T6 & T8 by 21%, 49% and 81% respectively” fits well with the statistical results.

Lines 342-343, please explain more details of the results first.

Response >

The authors have explained the results for root Si content in detail as asked.

  1. Lines 351-367

Lines 353-354, please explain the results of T5, T7

Response >

The results of T5 and T7 are now included in the text. Additionally, the Result for effect of Si supplementation on H202 content was also re-written according to the statistical results.

Lines 358-360, what is about the result of T8, please explain.  

Response >

Result for T8 has been added as suggested by the reviewer.

  1. Lines 368-399

Figure 6A, please check the letter on bars for a, b, c, d, e (T3= de?, T5 = bcd?, T7 = e?)

Response >

The authors agree with the reviewer’s suggestion. Authors have re-analyzed the data for statistical significance and have revised the figure 6A.

Figure 6A, please define the unit of enzyme activity in the methods for U mg protein/minute.

Response >

The authors have defined the unit of enzyme activity in the method section.

Lines 370-371. “With the exception of T3” what does the author mean?

Response >

It is usually found that salinity stress reduces the antioxidant enzyme activity in plants. However, in our case with 10mM NaCl stress we observed an increase in the antioxidant enzyme activities in T3. Thus, we have written the phrase “With the exception of T3” in order to convey this observation.

Lines 371-372, the authors wrote “Si supplementation was however seen to enhance the activity of SOD    by 185% and 101% in T6 and T8 respectively when compared to T5 and T7 respectively.”, but the results showed T5 and T6 are not significantly different, please consider.

Response >

Since the authors have re-analyzed the statistical significance, now the line “Si supplementation was however seen to enhance the activity of SOD    by 185% and 101% in T6 and T8 respectively when compared to T5 and T7 respectively” fits well in the context.

 Lines 378-379, the results are not significantly different, please consider (Figure 6B).

Response >

The authors have re-analyzed the data for statistical significance across various treatment groups and have revised the figure 6B. And have added detail explanation of results with respect to statistical analysis.

 Lines 382-385, the results are not significantly different, please consider (Figure 6C).

Response >

The authors have re-analyzed the data for statistical significance across various treatment groups and have revised the figure 6C. And have added detail explanation of the results.

  1. Lines 441-446, can Si supplementation positively influence RuBP in every salinity treatment, please provide data.

Response >

It depends on the plants/crops whether they are sensitive or susaptible towards salinity stress. There are several information in NCBI available how salinity can reflect the RubP

  1. Lines 535-551

 Figure 10 is not eight treatments, it is only four, please correct.

Response >

The error has been corrected as suggested by reviewer.

Figure 10, it is good to use T1, T2….., not M1, M2,…

Response > We prefer to use M1 as mungbean treatment 1……..so on as already we have provided the indication of them below it so it will be easy for readers to depict what M1-M4 stands for

  1. Expression of Si-transporter and salt responsive genes

Lines 536-543, the authors wrote “The expression of Lsi2 and Lsi3 genes was increased in salinity and Si supplemented groups (Figure 10A, 10B and 10C). However, the results show no significant different between NaCl and NaCl+Si. So, I disagree with the authors as it is the key message for the reader.

Moreover, Si application also did not significantly change SOS1, 2, and 3 as compared between NaCl and NaCl+Si.

Response >

We have re done the statistical analysis using SAS software, previously we have all together performed statistical analysis for all genes, we reperformed separately now for each gene and now it is evident form the data that there is a significant difference among all treatments.

Discussion

  1. Lines 576-579, please consider, see my comments for results

Response >

The comments of the reviewer from results section for soluble protein and soluble sugar content was followed and the lines mentioned were re-written in accordance with the revised results. Please check.

  1. Lines 587-589, please consider, see my comments for results

Response >

The comments of the reviewer from results section for phenolic content was followed and the lines mentioned were re-written in accordance with the revised results.

  1. Lines 604-608, please consider, see my comments for results

Response >

The comments of the reviewer from results for NR activity in leaves and roots were followed and the lines mentioned above were re-written in accordance with the revised results.

  1. Lines 616-618, please consider, see my comments for results. Also it is difficult to conclude that Si in leaves is greater than in root as the authors did not statistically analyse, I suggest analyse them.

Response >

The authors have re-analyzed the results for statistical significance and have revised the figures. In this context, the authors feel the discussion written fit well with the revised statistical results for Si content in leaves and roots. Based upon the significant indifference between T5 and T6 (root Si content) it can be said that Si content in leaves is greater than that in roots.

  1. Lines 630-632, please consider, see my comments for results.

Response >

The discussion for H2O2 and O2- content have been re-written by following the comments of the reviewer in the result section for the same.

  1. Lines 763-766, please consider, see my comments for results. (What are VrLis1, VrLis2, VrLis3?, They are not presented in Figure 10).   

Response >

We have corrected it, it is not vr since we have not sequenced them. Now changed to Lsi1, Lsi2, and Lsi3

  1. Lines 784-787, please consider, see my comments for results.

Response >

Authors have reanalyzed the statistical analysis, please see our response

  1. Lines 798-800, I disagree, please see my comments for results

Response >

Authors have reanalyzed the statistical analysis, please see our response

Conclusion

      Too long, please concise, and consider my comments for results and discussion.  

Response >

The Authors have re-written the conclusion such that the reviewer’s comments and suggestion regarding various results are incorporated.

      Abstract

       Please rewrite with more focus of statical analysis results

Response >

The authors have modified the abstract as per the reviewer’s instruction keeping in mind the text resonates with the statistical analysis results.

References

 Please follow the journal template as [number]

Response >

The Authors have revised the entire manuscript as per the journal template.

Reviewer 4 Report

Manuscript ID: agriculture-2484089

Manuscript Title: Physiological and Molecular Analysis Revealed the Role of Silicon in Modulating Salinity Stress in Mung Bean

This study aimed to understand the mechanism behind protection against excessive ROS, connect the various dots related to changes in protein expression under salinity stress and Si supplementation using LC-MS/MS, and examine the role of mRNA level regulation of Si-transporter genes and salt-responsive genes in mung bean under salinity stress and Si supplementation.

The manuscript is clear, technically correct. The English is overall ok, I have corrected some words/sentences that need to be corrected (but there are probably others that I did not notice in my revision). The structure of the manuscript can be said to conform to a generally acceptable form. This research is original primary research within the topics specified in the Purpose and Scope of the journal.

 (Please find the corrections in the annotated pdf)

Moderate editing of English language required

Author Response

#Reviewer 4

This study aimed to understand the mechanism behind protection against excessive ROS, connect the various dots related to changes in protein expression under salinity stress and Si supplementation using LC-MS/MS, and examine the role of mRNA level regulation of Si-transporter genes and salt-responsive genes in mung bean under salinity stress and Si supplementation.

The manuscript is clear, technically correct. The English is overall ok, I have corrected some words/sentences that need to be corrected (but there are probably others that I did not notice in my revision). The structure of the manuscript can be said to conform to a generally acceptable form. This research is original primary research within the topics specified in the Purpose and Scope of the journal.

Response >

The Authors thank the reviewer for their comments and response.

The Authors have further incorporated all the corrections in text that the reviewer has highlighted in the attached pdf. Kindly check.

Round 2

Reviewer 3 Report

1. Materials and methods

 Experimental design, when were seedlings treated with NaCl? Was it treated from germination to 10 days of age? Please clarify

 Response >

The Authors have treatment mung bean plants with NaCl after the plants had reached the vegetative stage. This is mentioned in the methodology section.

Reviewer:

I cannot find it, please indicate the age of plant when treated with NaCl in an article.

2. Results

Please improve qualities of Figures 4, 6A, B, C, 10.

Author Response

  1. Materials and methods

 Experimental design, when were seedlings treated with NaCl? Was it treated from germination to 10 days of age? Please clarify

 Response >

The Authors have treatment mung bean plants with NaCl after the plants had reached the vegetative stage. This is mentioned in the methodology section.

Reviewer:

I cannot find it, please indicate the age of plant when treated with NaCl in an article.

 Response >

The Authors have now included the age of plants when treated with NaCl and is highlighted in red in methodology section.

  1. Results

Please improve qualities of Figures 4, 6A, B, C, 10.

Response >

As per the Reviewer’s suggestion, the Authors have now improved the image quality of Figures 4, 6A, 6B, 6C and 10. The revised figure 10 may look a bit hazy, but if we zoom in we can see it having a clear image quality.
